

# Regional transport of aerosols from Northern India and its impact on boundary layer dynamics and air quality over Chennai, a coastal megacity in Southern India.

Saleem Ali[1], Chandan Sarangi[1*] and Sanjay Kumar Mehta[2]

[1]Department of Civil Engineering, Indian Institute of Technology Madras, Chennai, 600036, India

[2]Atmospheric Observations and Modelling Laboratory (AOML), Department of Physics and Nanotechnology, SRM Institute of Science and Technology, Kattankulathur, 603203, India

*Correspondence to: Chandan Sarangi (chandansarangi@civil.iitm.ac.in)

**Abstract.** Westerly driven regional transport of aerosols from the heavily polluted North India towards south-eastern India is a prevalent phenomenon during the winter season. Here, the regional aerosol transport events on the boundary layer dynamics and air quality over Chennai, a tropical South Asian megacity, are investigated. The long-term satellite data enables us to depict such regional transport events prolonged for a few days, accounting for ~10-13 per cent of the winter season. The occurrence of these regional transport events is increasing over time in southeastern India which are associated with relatively calmer conditions under anticyclonic wind circulation over north India extending to south India. The transported aerosol layer is generally located around ~1-3 km across the entire southeastern India, capped by the strong atmospheric temperature inversion. The regional aerosol/ haze transport significantly reduces the boundary layer height (ABL-H) by ~38% compared to clear sky conditions ( ~2-2.5 km). Consequently, an increase in $PM_{2.5}$ is observed to be ~30-35% in association with the strong heating aloft ABL (~1.2-2.5K), suppression of ABL-H and anticyclonic circulation over north India. This study provides robust observational evidence on the importance of regional transport of aerosols on air quality of downwind megacities and warrants more observational and modelling studies to constrain the inherent aerosol-induced effects on boundary layer dynamics.

## 1. Introduction

Atmospheric aerosols are pivotal in regulating Earth's climate systems by influencing radiation budget, cloud properties and biochemical cycles. Direct and indirect effects of aerosols on the radiation balance of the Earth-Atmosphere system are evident (Comstock and Sassen, 2001; Haywood and Boucher, 2000; Lohmann and Feichter, 2005; Satheesh and Krishnamoorthy, 2005; Yu et al., 2006) and it is believed to generate climate perturbations on a regional and global scale. Apart from the local generation, the long-range transport of aerosols from their sources can severely pollute a large area far from the apportionment and it is mainly influenced by the atmospheric circulation and aerosol lifetime. Although local emissions contribute mainly to hazy episodes in megacities, it can also be influenced by regional pollutant transports (Ma et al., 2020; Mhawish et al., 2022). Such hazy events can cause severe air pollution, adversely affecting public health. Prolonged haze events and associated high $PM_{2.5}$ loading have frequently been reported over South Asia and China during recent autumn and winter seasons (Qin et al., 2016; Yang et al., 2020; Zhang et al., 2021a). The significant factors influencing such hazy events were attributed to stable synoptic conditions with weak surface winds and low Atmospheric Boundary



Layer (ABL) height (Wang et al., 2014) along with the regional aerosol transport and ABL interaction (Zhang et
al., 2015).

38         Such transported aerosol layers, stratified above the ABL, can significantly affect the surface energy
balance and ABL dynamics owing to their interaction with incoming solar radiation (Ding et al., 2016; Ma et al.,
2020). Depending on the dominant aerosol species, the net impact of these layers could be absorbing or scattering
of incoming solar radiation. In either case, the presence of this transported aerosol layer can induce cooling at
altitudes below the layer and warming around and above the altitudes where they are located. Simultaneously,
near-surface accumulation of absorption aerosol concentration (under a shallower boundary layer) can lead to
lower atmosphere warming and surface cooling. Thus, a series of thermodynamical effects can ensue disrupting
stability and enhancing the upward transport of heat and aerosol through turbulent motion (Barbaro et al., 2014;
Huang et al., 2018). In continuation, previous studies found the role of aerosol on the suppression of ABL
development through their relative heating and cooling in the upper atmosphere and surface, respectively (Liu et
al., 2019; Petäjä et al., 2016; Wang et al., 2019b, 2020, 2018; Wilcox et al., 2016; Zhao et al., 2019; Zou et al.,
49    2017)

50         Hence, understanding and characterising the regional transport of aerosols on the ABL structure and air
quality are complex. There are studies signifying the role of aerosols on the boundary layer dynamics (Aruna et
al., 2013; Huang et al., 2018; Ma et al., 2022; Miao and Liu, 2019; Raatikainen et al., 2014); however, most of
them are based on the modelling framework, and observational evidence is scarce. This study aims to delineate,
for the first time, the effects of transported aerosols from north India towards the southern part of the Indian
peninsula on the boundary layer dynamics and hence the pollution dispersion using collocated high-resolution
lidar, radiosondes, surface weather observations along with space-based observatories.

57         The Indo-Gangetic Plains (IGP), the densely populated and growing economy of the Indian subcontinent,
experiences high aerosol loading both around the surface and in the vertical column during the winter season
attributed to the wide range of anthropogenic activities ranging from biomass, fossil fuel burning and agricultural
activities (Prasad et al., 2006; Ramanathan and Ramana, 2005; Tripathi et al., 2006). The prevalence of a high-
pressure system over the central Indian landmass, especially during the winter seasons (December to March),
generates a persistent northeasterly offshore flow (Krishnamurti et al., 1998). It provides a pathway for
transporting aerosols from continental areas into the otherwise pristine ocean, covering thousands of kilometres
in less than ten days (Krishnamurti et al., 1998; Rajeev et al., 2000). As such, pollutants from North India can get
transported to the Bay of Bengal and then towards South India under the influence of prevalent strong convection
and anticyclonic cyclonic circulation formation over the northwest of the Bay of Bengal (Prijith et al., 2016;
Rajeevan and Srinivasan, 2000). Such transboundary transport of pollutants is evident in widespread pollution
over the southern Indian peninsula (Ananthavel et al., 2021b; Kant et al., 2023; Mehta et al., 2023; Mhawish et
al., 2022; Ratnam et al., 2018; Thomas et al., 2021). There is a campaign-based investigation held over the Indian
Ocean, e.g., the Indian Ocean Experiment (INDOEX) (Ramanathan et al., 1995) to investigate the characteristics
of transported aerosols towards the Indian Ocean and the Arabian Sea (Chester et al., 1991; Prodi et al., 1983;
Savoie et al., 1989). The studies revealed that the transported aerosol predominantly consists of black carbon,
organics, sulfate, nitrate, ammonia, sea salt, and mineral dust (Ramanathan et al., 2001). An increase in the aerosol
loading in the free troposphere reduces the amount of incoming solar radiation reaching the surface, thus causing



dimming while warming the mid and upper troposphere and cooling the surface (Dipu et al., 2013; Sarangi et al.,
2018). On the other hand, they significantly alter the atmosphere's underlying thermodynamics, leading to
modifying the boundary layer structure. Hence, it is essential to characterise such transports, especially their
occurrence characteristics and the nature of the aerosols present. However, observational evidence on such
transboundary aerosol transports, their frequency of occurrences, their impact on the ABL development and the
regional pollution maintenance have not been attempted yet; this study primarily focuses on unravelling such
aspects.
Here, long-term satellite observations from Moderate Resolution Spectroradiometer (MODIS) and
Cloud-Aerosol Lidar and Infrared Pathfinder Satellite Observatory (CALIPSO) are used to understand better and
characterise the spatiotemporal variability in long-range regional transport of aerosols from North India to central-
southern India during the winter season. Further, we have also used collocated observations of Micro Pulse Lidar,
Radiosonde, surface weather and surface $PM_{2.5}$ measurements over Chennai to (i) investigate the widespread
haziness over Chennai due to these regional aerosol transport episodes and (ii) quantify the associated changes in
the surface meteorology, ABL height (ABL-H) and surface $PM_{2.5}$ distributions. Section 2 describes the datasets
used, followed by the methodology for composite analysis of aerosols during the Regional Transport Episodes
(RTE) days and clear days. Further, results and discussion are provided in section 4 and the conclusion in section

91  5.

**2. Dataset and Methodology**
***Space-based observations***
MODIS on board the polar orbiting sun-synchronous satellites (Terra and Aqua) is utilised to estimate
the aerosol optical depth (AOD) information at 550 nm. The MODIS measures radiance at 36 spectral bands in
the visible to thermal IR spectral range of 0.41-14 μm (Kaufman et al., 1997). Within the spectral range, 7 bands
are dedicated for aerosol measurement having a spatial resolution of 250m/500m. Owing to its large spatial swath
(2330 km), MODIS is capable of observing the entire globe in a single day during two different times, i.e., at
01:30 AM/PM (Aqua) and 10:30 AM/PM (Terra) local time, which crosses the equator. We used the current
version of Multiangle Implementation of Atmospheric Correction (MAIAC), which retrieves the AOD over land
at 1 km resolution (Lyapustin et al., 2011b, 2011a), between December and March during 2015-2024 in this work.
In addition, the space-based lidar observation, Cloud-Aerosol Lidar with Orthogonal Polarization
(CALIOP, (Winker et al., 2009; Young et al., 2013) onboard CALIPSO is utilised to understand the vertical
variation of aerosol extinction profiles. The level 2, 5 km (horizontal averaged) aerosol profile (AProf) at 532 nm
during December – March between 2015 and 2024, segregated during the RTE and clear days, are used. The
CALIPSO crosses the equator ~01:30 AM/PM; however, only the night passes (~01:30 AM) are used for the
present study owing to the better signal-to-noise ratio.
***In situ observations***
The Micro Pulse Lidar (MPL), an elastic backscatter dual-polarization lidar of Droplet Measurement Techniques
(DMT, USA), is located at the premise of SRM IST (45m above mean sea level). The instrument is set up at the
Atmospheric Observation and Modelling Laboratory (AOML, 40m above the ground level), at a total height of

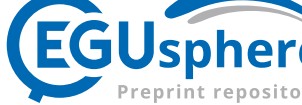

85m above mean sea level. The Normalised Relative Backscatter (NRB) is primarily utilised to retrieve total
attenuated aerosol extinction and determine ABL-H. Details on site description and technical specifications about
the MPL (Ali et al., 2022), retrieval of extinction coefficient and AOD (Ananthavel et al., 2021a, 2021b) and
ABL-H estimation (Kakkanattu et al., 2023; Reddy et al., 2021a) are provided in references.
Upper air and surface weather information used in this study are obtained from India Meteorological
Department (IMD) Chennai, from the sounding located at Meenambakkam (13.0N,80.06E, 16m above MSL),
about 20.13 km northeast of SRM IST, Kattankulathur and Karaikal (10.9N,79.8E, 6.9m above MSL). The
radiosonde data archived at 05:30 LT between December-January 2015 and 2024 are used to interpret the
meteorological conditions during the aerosol transport periods and ABL-H determination. The gradient of
potential temperature is utilised for the ABL-H determination (Mehta et al., 2017), where it is the altitude of the
maximum potential temperature gradient at the lower troposphere.
Hourly $PM_{2.5}$ measurements are routinely made at the U.S. Embassy and Consulate, Chennai using a beta
attenuation monitor (San Martini et al., 2015). The dataset within the study period is obtained from AirNow
(http://www.airnow.gov). The $PM_{2.5}$ observations from the U.S. Embassy are validated and in good agreement
with other observations (Jiang et al., 2015; Mukherjee and Toohey, 2016). The datasets are used to investigate the
distribution of surface pollution during the haze transport from IGP to Chennai.
***MERRA2 Reanalysis Data Products***
The Modern Era Retrospective analysis for Research and Application, Version 2 (MERRA2) is employed
to understand the spatial variation of Total Aerosol Extinction (TAE) and wind parameters (U, V, Speed). The
MERRA2 reanalysis product provided by NASA's Global Modeling and Assimilation Office (GMAO) is
available at 0.5° x 0.625° spatial resolution (Gelaro et al., 2017). MERRA2 simulates five aerosol species,
including sulfate, black carbon, dust, organic carbon and sea salt, with Goddard Chemistry, Aerosol, Radiation
and Transport (GOCART) model and their simulated properties are found to be robust (Randles et al., 2017).
**Methodology**
***Characterisation of RTE and Clear Sky days***
The RTE days are the days with significant aerosol transport from North India to the peninsular regions,
inducing a widespread haziness over the areas. Such aerosol transports are identified from the spatial variation of
AOD (MODIS), such that the AOD often exceeds 0.7 over the East Coast and nearby regions. Note that only days
showing a visible transport from the IGP (See supplementary figure Fig. S1 for a reference) towards the south
Indian peninsula are considered for the analysis. Conversely, there are days without any such transports;
occurrences of such days often promote visibility and reduce the AOD significantly (less than 0.3) and termed
them as 'clear' days. Note that days obscured with clouds, if any, are excluded from the entire segregation process
of both RTE and clear days.
Further, CALIPSO swaths available during the RTE and clear days are segregated over the eastern coast
within ±5° longitudes to study the spatial variation of extinction profiles pertinent to the corresponding events in
addition to the MERRA2 (spatially across the Indian subcontinent) Total Aerosol Extinction (TAE) and wind
products. Moreover, MPL (vertical profiles of extinction coefficient) and $PM_{2.5}$ measurements available during





RTE and clear days are segregated to infer the ABL-H variability and pollution concentration. Details of the
sample available for the analysis are presented in Supplementary Table 1.
**3. Results and Discussion**
**3.1 Occurrence of RTE and clear-sky days**
The climatological mean spatial distribution of columnar AOD over the Indian subcontinent from
RTE and clear days are compared in Fig. 1a and 1b. The significance of regional aerosol transport from north
India to south during these months is evident as the composite mean AOD over the entire central and southeastern
India during RTE days is 0.42±0.08 (Fig. 1a). The composite mean of AOD during clear days (Fig. 1b) is
substantially lower (0.23±0.06) over the region; however, as expected, it was greater than 0.7 over the North India
region. The RTE and clear days are observed to be 119 and 70 days, respectively, between 2015 and 2024. As
discussed earlier, the RTE shows an enhancement of more than 0.7 over the eastern coast of India compared to
the clear days.
The spatial distribution of aerosols during the RTE and clear days is further confirmed using the
MERRA 2 reanalysis products. The average of the total aerosol extinction (TAE) obtained from MERRA 2
(equivalent to columnar AOD at 550 nm), superimposed with the wind vectors, for the same composite of RTE
and clear-sky days are compared (Figure 1b-c). The analysis shows a similar pattern as observed from AOD
distribution from satellite data; however, variation in the magnitudes is present. Notably, the TAE exceeds 60%
on RTE days across the East Coast and nearby regions compared to clear days. During the winter and pre-monsoon
season, the westerlies wind system is prominent over Northern India and IGP in the lower troposphere prevalent
to the high-pressure system generation over central India (Krishnamurti et al., 1998). This flow manifests into a
northerly wind as it enters the Bay of Bengal and eventually merges into the easterly circulation (prevalent around
the tropics) as the wind enters back into the southern Peninsula, south of 20$^o$N. The fluctuations in this wind
circulation and the intensity of the high-pressure system induce hazy or clear-sky days occurrences. We checked
for any diurnal pattern in the TAE. However, such variations are negligible, pointing to the long-standing duration
of such events. Hence, we further examined the endurance of RTE days. Normally, the RTE events prolong for
days; persistence of such events for more than a day is observed to be ~53% of total observation, while 21% times
it prolonged for more than 4 days
Duration of RTE episodes, in general, can vary from one day to 4-6 days. Fig. 1e provides an
overview of such events' endurance between 2015 and 2024. Here, the number of consecutive RTE days
occurrence (N) is estimated every year, and they are further segregated into distinct day duration bins; further, N
is multiplied by the respective day duration to obtain the weighted count. The endurance of such events is colour-
coded as black, magenta and red colours for single days, continuous for 2-4 days, and lasting for more than 4 days,
respectively. Overall, endurance dominates the days exceedingly more than a day of RTE event. On average, the
RTE which occurred over the south-eastern coast box for consecutive 2-4 days is the highest (as shown in Figure
1e), and such events show an increasing pattern. Notably, the year 2022 experienced a 12-day consecutive haziness
between the 20$^{th}$ and 31$^{st}$ of March. Fig.1e also suggests an increasing trend in the overall occurrence of RTE days.
Endurance of such hazy periods can result in significant consequences on the air quality and boundary layer
dynamics. Note that, although a day between consecutive days is absent, e.g., the second day of three consecutive



days of an event, it is counted in this statistics to overcome the instrument limitation. It is worth noting that the
easterly wind speed across the southern BoB is stronger during clear days than on RTE days. As such, the
difference between the RTE and clear-day wind speed composites is that the wind speed during RTE days across
the entire eastern half of the Indian subcontinent is weaker than the clear days (Fig 1f). The reduced wind speed
is expected to promote the accumulation of aerosols and induce greater AOD over the south-eastern coast and the
southern Indian peninsula, hence promoting the RTE.
The CALIOP observed mean vertical distribution of the aerosol extinction over the south-eastern coastal
region (bounded box in the dotted line in Fig.1a and b) during the RTE days and clear-sky days is shown in Fig.2a
and b, respectively. As expected, there is a distinct decreasing gradient in aerosol extinction values between surface
and 2 km altitude as we move from IGP in the north to southern peninsular coastal India during both RTE and
clear days. More interestingly, high values of aerosol extinction (> 0.15) are discernible up to 5 km during RTE
days over the region south of 20ºN during RTE days, while the same is confined to altitudes less than 1.5 km
during clear-sky days. Also, over region north of 20ºN, a relative increase in the extinction within ~2-4 km can be
observed on RTE days relative to clear days. Several studies suggest the extension of aerosol layers more than 3
km. For instance, dust aerosols can reach up to 4-7 km during sandstorm episodes, as observed over the
northwestern Tibet Plateau (Huang et al., 2007) and 3-5 km in urban Beijing (Guo et al., 2014). Using both
CALIPSO and ground-based lidar measurements, (Qin et al., 2016) shows the transboundary transport of aerosols
in China, having an aerosol layer depth of up to 3 km, during a haze event that occurred in the winter of 2015.
Temporal changes in the vertical characteristics of aerosol extinction, ABL and TAL during an RTE event
followed by clear days, as observed by MPL, is provided in Fig.2b. Such that the vertical distribution of total
attenuated aerosol extinction between 23-29 January 2018 which includes RTE episodes (24-27 January) and
clear-sky conditions (23 and 28-29 January) as shown in Figure 2b. The figure also includes the vertical
temperature profiles (T, radiosonde) and the ABL-H (MPL). The occurrence of the transported aerosol layer (TAL)
can be seen above the ABL during the RTE periods and persisted for ~ 3-4 days. The top of the TAL (red dotted
line) was identified as the prominent peak above the ABL, determined from the NRB gradient (Kakkanattu et al.,
2023; Mehta et al., 2023). The TAL was observed initially at ~2.5 km at ~06:00 LT (or IST) on 24 January 2018,
which gradually reduced to ~1.5 km and merged with the ABL at 09:00 LT on 28 January 2018. The temporal
variation of background surface meteorology, including surface T, wind speed (WS), PM2.5 and the AOD, are
provided in the Supplementary Figure Fig.S2. A significant increase in the columnar AOD between ~0.4 and 0.8
is observed during the hazy events. However, it maintains ~0.2-0.3 during the clear days. It is also worth noting
that the AOD within the ABL (integrated extinction within surface and ABL-H) decreases from ~0.4 to less than
0.2 during the RTE period, suggesting a dominant presence of TAL above the ABL. Such aerosol accumulations
are followed by strong upper atmospheric warming and surface cooling (Liu et al., 2019; Zhao et al., 2019). As
discussed earlier, the atmosphere at the altitude, where the TAL presents, is observed to be warming. However,
contrary to the earlier findings, the surface temperature is also observed to be warming. The effect of aerosols on
the radiative forcing is mainly dominated by their microphysical characteristics, such as their absorption or
scattering nature. Studies on such aspects are relied on in the future scope.
Interestingly, the ABL-H also decreased from ~1.4 km to ~0.3 km (~78% reduction) between 24 and 25
January 2018. The climatological ABL-H is observed to be 1.3±0.8 km and 1.8±0.8 km during the RTE and clear



days, respectively, exhibiting an overall reduction of ~38%, indicating the role of RTE events in altering the
boundary layer dynamics. The suppression of the ABL followed by accumulation of absorptive aerosol in the
upper ABL has been investigated earlier, especially through numerical simulations (Ding et al., 2013, 2016; Zhao
et al., 2019); however, observational evidence is scarce. (Barbaro et al., 2014) suggests that a drop in the ABL
from 1.4 km to 0.9 km (~35% reduction), through sensitivity experiments. Similarly, (Wang et al., 2015) suggest
that the stable stratification of the atmosphere above the ABL through the significant warming by absorbing
aerosols contributed to a decrease of ABL-H by 33%. Observational studies over China suggest that the occurrence
of elevated aerosol layers has induced suppression of ABL-H from 1.27 km to 0.78 km (~38% reduction) hence
lifting the surface pollution level to 118% (Wang et al., 2018). (Zhang et al., 2021b) reports a reduction in the
ABL-H from 1.09 km to 0.48 (~60% reduction) during intense haze episodes over China. Unfortunately,
observational evidence of such aspects is not attempted over peninsular India, especially when transport events
occur.   The temperature profiles (Fig.2c) show a strong inversion at the altitude of ABL and the top of the TAL
during the hazy days, attributed to the aerosol heating. This contention further suggests that the presence of strong
temperature inversion over the ABL, especially during the dry season, can be accounted for the presence of TAL
followed by widespread haziness, increased temperature and pollution.

241         The study further extends to the diurnal changes in the aerosol extinction. Fig.2d shows the diurnal
variation of difference in total aerosol extinction during the available RTE (10 days) and clear (6 days) day cases
within 2018 and 2023 obtained from MPL. Notably, a strong enhancement in the extinction coefficient during
RTE days is discernible, especially above the ABL with a thickness of ~1.5-2 km; however, an overall reduction
is observed within the ABL except during the forenoon hours. The accumulation of such aerosols can strongly
impact the evolution of ABL. Fig.S3 shows the scatter plot between the ABL-H and AOD observed above the
ABL (AOD$_{aloft}$, integrated extinction above ABL-H), showing a strong inverse relation with a statistically
significant correlation of -0.42. It points out that the observed reduction in the ABL-H prevalent to the RTE is
linked to the optical depth of the aerosol aloft. Additionally, the enhancement in the aerosol extinction in the
atmosphere (both the scattering and absorption of sunlight) can significantly reduce the short-wave radiation of
incoming solar radiation, hence reducing the surface heat flux and development of ABL, resulting in the inhibition
of ABL-H (Ding et al., 2013; Li et al., 2017; Petäjä et al., 2016; Quan et al., 2013). On the other hand, absorbing
aerosols can heat the atmosphere, leading to temperature inversion (Xu et al., 2019).  It is also worth noting that
the surface extinction during the RTE days increased during the forenoon hours. Such enhancement can
significantly impact the PM$_{2.5}$ dispersion during those periods. Inhibition of ABL development by the strong
absorbing aerosols in the upper ABL can trigger high concentrations of pollution at the surface (Ding et al., 2016;
Petäjä et al., 2016; Zou et al., 2017) and such aspects are discussed in Sec. 3.3.

**3.2 Meteorological conditions during RTE and clear days**

260         To understand the effect of the transported aerosols and their vertical extent on the background
metrological conditions, we also analysed the vertical profiles of the wind speed (WS), relative humidity (RH)
and temperature (T) obtained from radiosonde observations over the two coastal stations; Chennai (13.0˚N,
80.0˚E) and Karaikal (10.92˚N, 79.83˚E), located in the east coast of the Indian peninsula where Karaikal is



departed by ~240 km from Chennai. The average profiles with standard error obtained for the RTE (red) and clear
(blue) categories are shown separately for Chennai and Karaikal, the first and second raw panels. The difference
between the RTE and clear days (RTE-Clear) is shown on the top axis as dashed lines. Although the wind direction
is north easterlies up to 3 km in both stations, the WS varies during the RTE and clear days. The WS up to 3 km
is almost similar to that of Chennai and Karaikal during the RTE days except at the surface. However, it exceeded
by ~2 m/s during clear days over both stations. The WS is observed to get stronger above ~1.5 km during the RTE
days. Such an enhancement in the WS can favour the transport from IGP and accumulate over the boundary layer.
However, the overall WS during RTE days remains calmer than on clear days. The effect of the accumulated
aerosol on the boundary layer processes majorly depends on the aerosol characteristics. For instance, the
absorption aerosol concentration above the ABL heats the thermal inversion layer and strongly suppresses the
ABL development (dome effect) (Ma et al., 2020). The shallow ABL further promote severe hazy episodes (Quan
et al., 2014; Ye et al., 2016). On the other hand, the radiative impact of the aerosols and their feedback on the haze
can further intensify the pollution (Ding et al., 2016; Yang et al., 2016).
A rapid decline in the RH from ~80% to 50% is observed during RTE days between the surface and 1.5
km (Fig.3b and e). However, it gradually decreases from ~80% to 40% during the clear days. In general, the
persistence of such RH above the boundary layer is expected to enhance the hygroscopic growth of the transported
aerosol layer and thus increase their endurance period (Zhao et al., 2017). However, compared to clear days, it
observes a rapid decline in the RH up to ~20% during the RTE days till ~ 1 km. This contention further points to
the absence of the hygroscopic growth of aerosol above the ABL; rather, the pollutants are mostly transported,
favoured by the prevalent wind system. Notably, a strong increase in the temperature is observed within ~1.5 km
is during RTE days (Fig.3c). Although Karaikal experienced an enhancement of ~1.2 K, Chennai peaks to ~2.5
K. Such increment in the T suggests the radiative effects of transported aerosol on the boundary layer. This
phenomenon also suggests that aerosol-induced warming at the lower troposphere not only increases the
temperature at the altitude where aerosol lies but also modifies the overall temperature profiles of the lower
atmosphere. Observational study by (Huang et al., 2018) over Northern China region shows a significant heating
in the upper ABL, where the aerosol accumulation is more, with a maximum temperature change of ~0.7°C on
average. Recent studies suggest that the occurrence of the aerosol layer in lower troposphere warming while
inducing a strong inhibition layer, and it may further promote extreme precipitation events (Dagan and Eytan,
292  2024).

Considering Chennai and Karaikal stations, although the WS and RH variation during the RTE and clear
days are similar, the observed change in the temperature varies significantly regardless of the limited distance
between the two stations (~2° meridionally). It suggests the spatial inhomogeneity in the TAL distribution. A
reduction in the AOD distribution by ~20-22% (from~ 0.9 to 0.7) (Fig.1a) between Chennai and Karaikal
decreased ~1K or 35-40% (from ~2.5 to 1.5) of regional temperature. Additionally, the surface is observed to be
less warm over Karaikal compared to Chennai during the RTE days.
**3.3 Effect of transported aerosols on the boundary layer and air quality**
As mentioned earlier, the occurrence of RTE induces a strong warming around the top of ABL. Such
warming is expected to change the vertical temperature stratification and thus impede the development of ABL,



commonly termed as "dome effect" of aerosols (Ding et al., 2016). Figure 4a examines the difference between the
temperature between 1.5 km and surface (ΔT) during the RTE and clear cases. Although clear cases distribution
peak maximizes around -8°C, the RTE skews towards -4°C, showing the relative role of transported aerosol on
the temperature stratification and inducing the "dome effect". The relative suppression of ABL development under
the influence of TAL is shown in Figure 4b.
Figure 4b shows the probability distribution of ABL-H estimated from the radiosonde profiles over
Chennai during both the RTE and clear days. Interestingly, the PDF during RTE days peaks around ~0.5 km while
it is 2-2.5 km during the clear days. Skewness of ABL-H distribution towards the lower altitudes, especially during
the RTE days, suggests the relative role of TAL on the suppression of ABL. Such suppression in the ABL-H was
dominant during the afternoon hours, as observed from the MPL (see Fig.2b). The development of ABL during
daytime is mainly dominated by convective processes (Garratt, 1994; Stull, 1988); however, the formation of TAL
suppresses such development, hence reducing the ABL-H. On the other hand, the complex radiative interaction
between the incoming solar radiation with the TAL, especially the surface dimming and inducement of the "dome
effect", can also affect the development of ABL (Guo et al., 2017; Petäjä et al., 2016). The solar dimming due to
the presence of TAL can block the solar radiation reaching the surface, resulting in the overall dimming in the
ground surface, weakening the surface flux, perturbing the convective process and suppressing the ABL
development. Enhancement in the suppression of ABL-H distribution during the afternoon hours can also be
attributed to the thermal internal boundary layer formation, where the transport of pollutants towards the land
from BoB under the influence of sea breeze (Reddy et al., 2021b). However, this contention requires further
investigation with large samples. As mentioned earlier, the suppression of ABL-H in the presence of TAL could
substantially enhance near-surface haze pollution (Petäjä et al., 2016; Sun et al., 2024; Zou et al., 2017).
The association of ABL-H and $PM_{2.5}$ is delineated using collocated observations of MPL and $PM_{2.5}$
measurements over Chennai. The RTE days where MPL is operational are only used for this analysis. Fig.4d
shows the scatter of normalised anomalies between $PM_{2.5}$ and ABL-H. The normalised anomalies are obtained by
subtracting the parameters from the climatological average (during the winter season) and further divided with
the same ($X\ Anom. = \frac{X - X_{Climatology}}{X_{Climatology}}$, where X = $PM_{2.5}$, ABL-H). Note that, average ABL-H during the winter
season is estimated with the NRB during 2018 and 2023 alone. However, it matches with the climatological ABL-
H estimated by (Reddy et al., 2021a) over Chennai. Interestingly, the normalised anomalies between $PM_{2.5}$ and
ABL-H are negatively related, with a statistically significant (>95% confidence) correlation of -0.38. It portrays
that, overall, a 50% reduction in the ABL-H contributed to ~100-150% increase in the surface $PM_{2.5}$
concentrations. Observational studies by (Su et al., 2020) show a nearly similar relationship during COVID-19 in
China; however, with a different correlation value over Beijing and Northern China, attributed to the
inhomogeneity in the spatial distribution of pollution.  It is to be noted that, the wind speed and direction are the
major influencing factors affecting the spatial distribution of aerosols. Hence, we further analysed the contribution
of WS and WD explicitly. The observed negative association becomes stronger when the WS exceeds more than
4 m/s (-0.46, N=40) and also when the WD from the northeast direction (-0.5, N=56). Such cases show the $PM_{2.5}$
aggravation to ~150-200% when the ABL-H suppressed by 50% (Fig.4d).  It suggests that RTE favoured by the
enhanced WS, directed from north India, influences the formation and maintenance of TAL over the southern



Indian regions, and it further suppresses the local boundary layer while increasing the surface pollution
concentration.
In general, an increased ABL height, as usually occurs during clear days, can result in acceleration of
surface wind speed and enhanced vertical movement (Xiang et al., 2019), promoting reduced surface pollution.
On the other hand, the occurrence of residual layer (RL) or stable boundary layer (SBL) complicates the pollution
dynamics at the surface (Yu et al., 2020). The observed anomalies in the surface pollution in the cases of clear
days can be attributed to the re-entrainment of pollutants in the RL into the mixed layer, leading to rapid change
in the surface pollution concentration (Shi et al., 2020; Yu et al., 2020). In addition, the presence of SBL, which
hinders the exchange of pollutants and energy between the surface and free atmosphere, potentially leads to higher
concentrations of pollutants in the atmosphere if they are not cleared otherwise (Shi et al., 2020), similar to the
RTE conditions.
Finally, the overall diurnal changes in $PM_{2.5}$ during the RTE and clear day composites are portrayed in
Fig.4d. As expected, the RTE days experience ~30-35% enhancement in $PM_{2.5}$ than clear days. The surface $PM_{2.5}$
increases during the early morning hours and maximizes at 08:00 LT. Although the clear-day composite shows a
gradual decrease from 50 $\mu g/m^3$ to 30 $\mu g/m^3$ between 08:00 and 15:00 LT, it rapidly decreases from 60 $\mu g/m^3$ to
32 $\mu g/m^3$ during RTE days. As depicted in Fig.2d, the enhancement in the aerosol extinction coefficient is minimal
during the afternoon hours during the RTE days. The observed variation in surface pollution is also linked with
the ABL-H. An increase in the ABL-H makes a pathway for the dispersion of air pollutants, hence reducing surface
pollution levels. In contrast, a suppressed ABL-H significantly affect the vertical dispersion, leading to higher
concentrations of pollutants near the surface (Wang et al., 2019a).
**4 Summary and Conclusion**
This paper presents the first observational evidence of the effect of transboundary transported aerosols on the
boundary layer dynamics and pollution dispersion over the east coast regions of peninsular India. The aerosol
transport from IGP towards south India is segregated from the spatial distribution of AOD using MODIS. Such
transboundary transports (referred to as RTE), mainly influenced by the anticyclonic circulation formation over
the northern BoB and prevalent north easterlies, contribute widespread haziness spatially and vertically over the
west coast of BoB and nearby regions.
The occurrence of RTE days generally prolongs for more than a day. The largest lapsed RTE were observed
during March 2022, which continued for 12 consecutive days. The widespread haziness over the northeastern
region of the Indian subcontinent induced by the occurrence of TAL, especially during the winter season, is mainly
favoured by the relative reduction of wind speed during the season near the surface. The reduced windspeed
declines the dispersion of pollutants over a large area in the southern Indian peninsula hence reducing any diurnal
variability in the aerosol distribution. On the other hand, although the overall WS declines during RTE days
compared to clear days, it shows some enhancement above ~1.5 km, promoting the formation and growth of TAL
aloft the ABL.
The TAL has a ~1-2 km thickness and occurs just above the ABL. A strong temperature inversion between
the top of the TAL and free troposphere is observed during the RTE periods, followed by a strong warming up to
~1-1.5°C where the TAL is present. Overall, the RTE and occurrence of TAL have suppressed the ABL-H by



~38%; however, for a typical RTE episode, temporal variation in the aerosol extinction characteristic suggests a
suppression of up to ~78 %. The occurrence and maintenance of TAL during the RTE are favoured by the strong
wind flow from north India, which majorly contributes to the reduction of the ABL-H and aggravation of surface
pollution. The aerosol "dome effect", as a result of the vertical temperature stratification due to the presence of
TAL, has induced the suppression of the ABL-H and increased the surface $PM_{2.5}$ by ~30-35% compared to the
clear days. This study elucidates the first qualitative investigation of the transboundary transport of aerosols over
the Indian peninsula and is a reference for emission policies over the eastern coasts, especially over Chennai and
the surrounding area. The analysis of the TAL is carried out by removing the cases of shallow clouds occurring
frequently during the study period, which we would like to pursue in a future study.
**Data Availability**
MODIS and MERRA2 data can be obtained from NASA Goddard Earth Sciences Data and Information Services
Center (GES DISC). CALIPSO data used in this study can be obtained directly from the website
https://eosweb.larc.nasa.gov/project/calipso/calipso_table. Radiosonde and surface data can be obtained from the
https://weather.uwyo.edu/upperair/sounding.html. The MPL data used in this study are not publicly available;
however, the data can be provided to the corresponding author upon request.
**Author contributions**.
SA was responsible for carrying out the investigation, writing, reviewing, data curation, and preparing the original
draft of the paper. CS is responsible for conceptualizing and supervising the study, carrying out the investigation,
writing, reviewing, and editing the paper. SKM is responsible for reviewing and editing the paper
**Competing interests.**
The contact author has declared that neither they nor their co-authors have any competing interests
**Acknowledgements**
SA thanks the Asia-Pacific Network for Global Change Research (APN) research grant (CRRP2022-08MY-
Sarangi) for supporting this work








**Figures.**

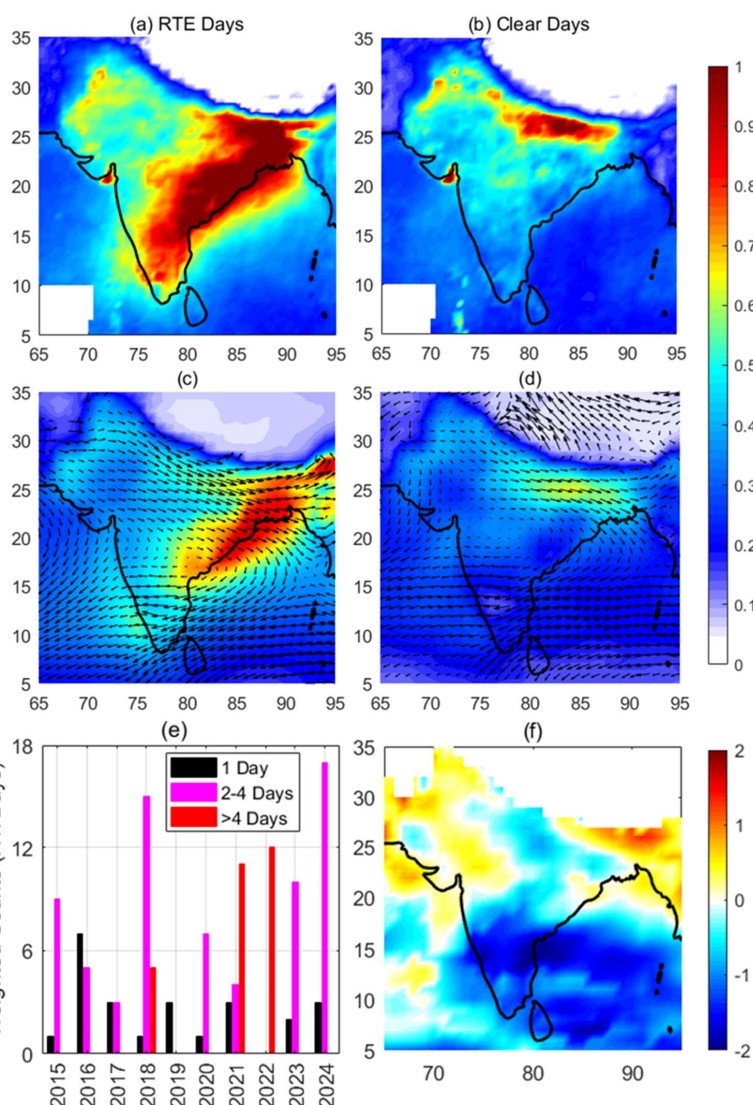


**Figure 1 Composite of the spatial distribution of AOD obtained from MODIS during (a) RTE and (b) clear**
**days between December and March during 2015-2024 and total aerosol extinction (TAE) from MERRA2**
**reanalysis dataset observed for the composite of (c) RTE and (d) clear days. Panel (e) shows the statistics of**
**occurrence frequency of RTE day periods segregated for 1 day (black), 2-4 days (magenta) and more than**
**4 days (red). (f) Difference in the wind speed between RTE and clear days at 850 hPa.**

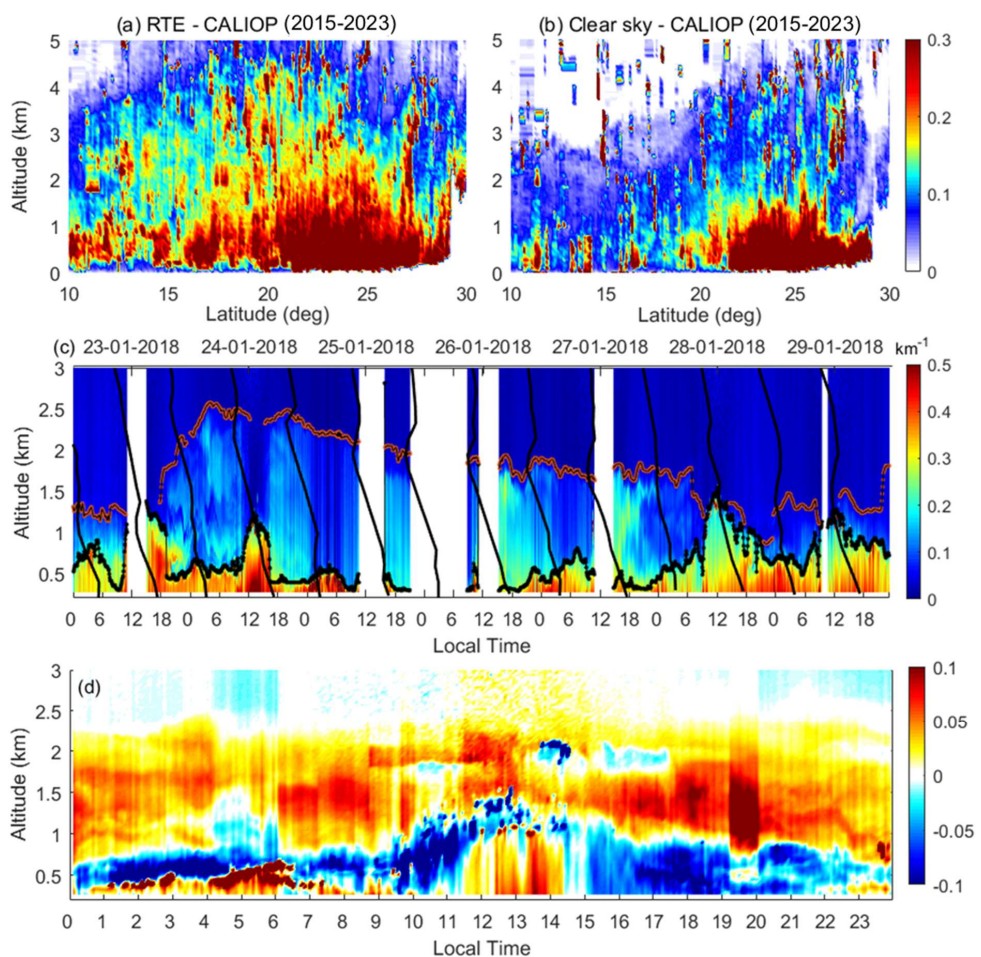

**Figure 2. The vertical distribution of the aerosol extinction coefficient from CALIOP during the (a) RTE and (b) clear days within ±4° longitude over the eastern coast of India between December and March 2015-2023. (c) Time-Altitude cross-section of the total attenuated extinction coefficient obtained from Micro Pulse Lidar (MPL) observation over Chennai (SRM IST) between 23 and 29 January 2018. The black line corresponds to the temperature profiles from radiosonde over IMD, Chennai. The black dotted line corresponds to the derived ABL-H, and the red dotted lines are the top of the transported aerosol layer (d) Temporal changes in the mean difference between the extinction coefficients during the available RTE and clear day composites (RTE – Clear) from MPL observations.**

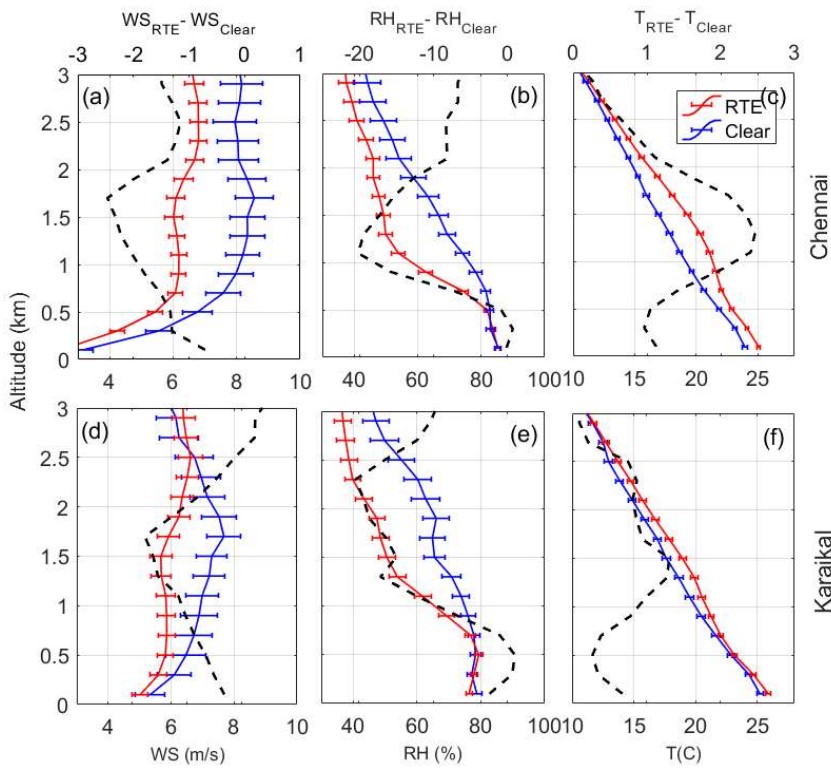

**Figure 3 Vertical variation of Wind Speed (WS), Relative Humidity (RH) and Temperature (T) over Chennai (a-c) and Karaikal (d-f) during RTE (red) and (b) clear days (blue). The difference in the RTE and clear days (RTE-Clear) are shown in dashed line (axes in top). The horizontal bars correspond to the standard errors.**

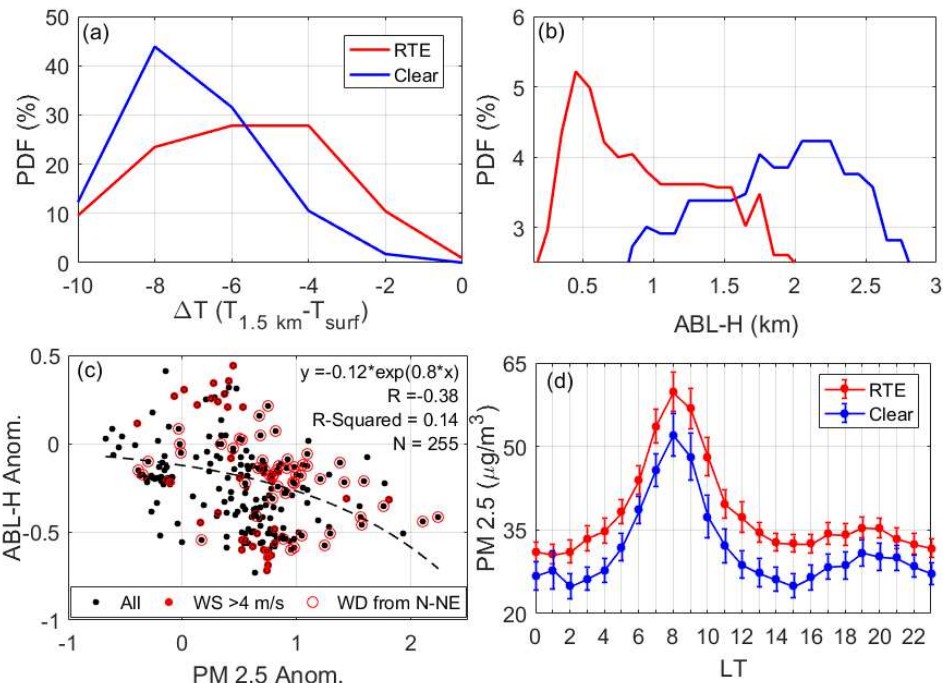

**Figure 4. Probability distribution (%) of (a) difference in temperature between 1.5 km and surface (ΔT; bin size=2K) and the (b) ABL-H obtained during climatological RTE (red) and clear (blue) cases (bin size=0.2km). (c) Scatter plot showing the normalized anomaly of PM 2.5 and ABL-H (obtained from MPL) during the RTE days as observed by MPL. The red scatters show the cases when wind speed (WS) is exceeds 4 m/s and red circles for the cases where wind directed from north to north east. The correlation coefficient, exponential fit (dashed line) equation, R² and number of samples also provided. (d) Diurnal variation of PM₂.₅ over Chennai (US Consulate, Chennai) during RTE and clear days.**



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
