# Peer review of "Regional transport of aerosols from Northern India and its"

_EGUsphere, 2024_

## Referee Comment (RC1)

**Overview/General Comment**

The manuscript entitled "Regional transport of aerosols from Northern India and its impact on boundary layer dynamics and air quality over Chennai, a coastal megacity in Southern India." by Ali et al. attempted to study the critical influence of aerosols on boundary layer dynamics. The research problem is highly significant and holds considerable importance within the scientific community. However, the results are very loosely presented with inadequate justification. The present analysis ranging from ABL identification to TAL transport, relevant interpretation requires greater depth and clarity to attempt the present problem. The estimation of ABL-H, ABL-AOD need to be revised in all the sections before presenting the analysis. Given these shortcomings, I believe substantial revisions are necessary. Therefore, I recommend resubmission, provided the authors address the following comments,

**Specific Comments**

Line no 105: The statement "during December – March between 2015 and 2024" is inaccurate as the CALIPSO mission ended on August 1, 2023. needs clarification on this statement.

Line no 153: The authors refer to "The climatological mean spatial distribution of columnar AOD." However, defining 2015 to 2024 as a climatological period is inappropriate. Please revise accordingly.

Line no 158-160: The statement, "The RTE and clear days are observed to be 119 and 70 days... RTE shows an enhancement of more than 0.7 over the eastern coast of India compared to clear days," requires clarification: (1) What latitude/longitude range was averaged to derive the value of 0.7? (2) Is the RTE enhancement consistently above 0.7 for all 119 days?

Line no 195-197: "decreasing gradient in aerosol extinction values between surface and 2 km" between RTE and Clear days. This observation is very important. However, it is also important the uncertainty associated with it, (i) author mentioned Fig 2a and 2b is a mean extinction between Dec and March 2015 to 2023 and the (ii) passes considered for the present study is ± 5 deg longitudes which include land and ocean. Considering that the following uncertainty arises,

December features strong ABL inversion (~1 km), while March inversions can extend to ~2.5 km. Similarly, oceanic passes may yield aerosol peaks near 1 km, whereas land passes could show concentrations up to 2 km. I recommend presenting RTE and clear-day analyses for individual months and separating land and ocean data to address these uncertainties.

Line no 210: "Can be seen above the ABL during the RTE periods" I recommend to show the 3 or 5 days backward trajectory analysis from surface upto 4 km during RTE and Clear sky days.

Line no 213 – 214: "temporal variation ....AOD" I have noticed several instance the ABL AOD towards Zero. This reflects the ABL estimation in figure 2c has been associated with uncertainty. For instance, on 27-01-2024 after 18 LT the ABL height and the first altitude bin are at the same altitude. In real atmosphere it never exists. This needs to be taken care in all the section of ABL height identification otherwise the claim of ABL height reduction, boundary layer dynamics may not be valid . It is also to be noted the entire manuscript depends on ABL identification to discuss the TAL and BL dynamics.

Line no 217: "..ABL (integrated extinction within surface and ABL-H) decreases from ~0.4 to less than 0.2.." re-estimation of ABL altitude, ABL-AOD needed.

Line no 220-221: "....the altitude, where the TAL presents, is observed to be warming...." "...contrary to the earlier findings, the surface temperature is also observed to be warming ..." it is confusing. 1) What analysis supports the warming conclusion? Temperature profiles typically decrease with height in Fig 2c. 2) Which earlier findings author mentioning? As similar how authors concluding the surface temperature warming? Is author mentioning the diurnal variation ? with respect to what it is warming ?

Line no 224: "..ABL-H also decreased from ~1.4 km to ~0.3 km.." I can see the ABL temperature inversion of at ~ 1.0 km during 25-01-2018 from the radiosonde observations.

Line 225: "The climatological ABL-H".. can be replaced to "The mean ABL-H.." use of climatological may not valid.

Line no 226-227: "RTE events in altering the boundary layer dynamics" it is loosely stated, I recommended authors should propose the mechanism, appropriate explanation with proper evidence.

Line no 237: "strong inversion at the altitude of ABL and the top of the TAL during the hazy days, attributed to the aerosol heating" the strong inversion layer is related to the large-scale seasonal variation Sinha et al (2013), Ganguly et al (2006). Interpretation needs to be revisited.

Line no 242-243: "RTE (10 days) and clear (6 days) day cases within 2018 and 2023 obtained from MPL". I recommend the author to present within a month or within the season of particular year to avoid the uncertainty related to inter and intra variability.

Line 249-255. "….significantly reduce the short-wave radiation of incoming solar radiation, hence reducing the surface heat flux and development of ABL…." The author's claim is not relevant from the present observation. I recommend perform flux direct observations or at least from reanalysis data to support this statement.

Line 259 – 298: Section 3.2. I did not understand why the author decided to present the meteorological observation between Chennai and Karaikal. When the TAL is from the IGP?. Spatial analysis, or data from stations along the transport pathway from IGP to Chennai, would be more appropriate.

Section 3.3. The analysis presented is insufficient to substantiate discussions on BL dynamics. Consider including spatial analyses from CALIPSO or other reanalysis datasets to strengthen this section.

**Minor Comments:**

Line 100: The statement "AOD over land at 1 km" mention including Ocean as well.

Line 146: The phrase "eastern coast within ±5° longitudes" is ambiguous. Clarify whether this refers to a specific latitude/longitude or the entire eastern coast. Define the latitude/longitude range considered in this study.

 Line 155: The authors state "during these months is evident," but do not specify which months are being referenced.

 Line 205: The acronym TAL should be introduced here, rather than in Line 209, for better readability and context.

 Lines 156–157: Specify the latitude/longitude range averaged to derive the AOD values for RTE days (0.42 ± 0.08) and clear-sky days (0.23 ± 0.06).

Line 239: Replace "during the dry season" with "during the winter season" for consistency and clarity.

---

## Author Comment (AC1)

Given the current state of the manuscript, I recommend substantial revisions and resubmission, addressing the following comments to align with the journal's standards for scientific quality, clarity, and rigor.

**We thank the reviewer for the recommendation as well as for proving constructive comments and suggestions to improve the manuscript further. We have addressed all your comments and our response is provided below.**

Comment: Mention the specific version of the CALIPSO data used in the current study. This is critical for understanding the availability and applicability of the data, especially since CALIPSO data is not available beyond 2020/2023, depending on the version.

**Reply: In this study, CALIPSO lidar level 2, 5 km standard aerosol profile product of version 4.51(CAL_LID_L2_05kmAPro-Standard-V4-51) available over the period between June 2006-June 2023. We used dataset between 2015 and 2023 for the present study. This information is provided in the revised manuscript.**

Comment: Line 105: It was mentioned that that CALIPSO data was used "during December–March between 2015 and 2024." However, the CALIPSO dataset does not extend beyond 2020/2023 (depending on the version). This needs clarification.

**Reply: We thank the reviewer for pointing out this typo, which we have corrected in the revised manuscript. CALIPSO datasets over the period between December and March during 2015-2023 is utilized in this study.**

Comment: Line 130 (U, V, and Speed): Mention "Speed," but it is unclear what this refers to in the context of winds (U, V components). Clarify whether this represents the resultant wind speed or another parameter.

**Reply: We have clarified it in the revised manuscript as Zonal wind (U), Meridional Wind (V) and resultant wind speed ($\sqrt{U^2 + V^2}$).**

Comment: On what basis it is identified that aerosols being transported from the Indo-Gangetic Plain (IGP) to Chennai? Was any pathway analysis (e.g., HYSPLIT trajectories, wind back-trajectory models, etc.) carried out to confirm the transport of aerosols?

**Reply: We thank the reviewer for this comment. The main basis for aerosol transport from the Indo-Gangetic Plain to the southern Peninsula is divergence associated with the anticyclonic circulations prevalent during the winter season. It provides a pathway for the transport of pollutants from IGP to the Bay of Bengal (BoB) and further inland over the southern Indian peninsula. We have included detailed back-trajectory analysis now in the revised manuscript. Although the wind system at 850 hPa (as shown in Fig.1c of revised manuscript) provides an**

estimate of such transport, we also utilized the HYSPLIT wind back trajectory model to better understand the pathway characteristics. Figure 1 e, f presents the number density of NOAA-HYSPLIT trajectory analysis for the RTE and clear days between surface (50m) and 4 km, illustrating the 5-day backward trajectories from Chennai, computed for every 6-hour interval. It also confirms that pollutants are predominantly transported across the BoB from the northern parts of India during RTE. In addition, there also observed transport from inland areas of the eastern coasts. On the other hand, the transport is predominantly from the nearby oceanic region during the clear days

[Figure]

*Figure 1. The number density of 5-day backward trajectories initiated from Chennai between surface and 4 km during (e) RTE and (f) Clear days.*

Comment: Elaborate more on how clear days and Regional Transport Events (RTE) are categorized, including specific criteria and thresholds.

[Figure]

*Figure 1.Composites of AOD observations from MODIS are shown separately for (a) RTE and (b) Clear days. The spatial region used to identify the RTE and Clear days are shown as a dotted box, named as eastern coastal box.*

Reply: The methods for the categorization of Regional Transport Events (RTE) and clear days are elaborated on in the revised manuscript. The RTE days are characterized by significant aerosol transport from northern India to the peninsular regions, resulting in widespread haziness

**across these areas. These aerosol transport events are identified based on the sudden temporal variation of aerosol optical depth (AOD) estimated from MODIS observations. Specifically, "RTE days" are defined as those days when the mean AOD exceeds 0.7 over the southeastern coastal box and adjacent regions, as illustrated in Fig. 1. In contrast, days with significantly reduced values of mean AOD (AOD < 0.3) over the spatial box are classified as "clear days".**

**Further, we performed back-trajectory analysis for all the identified days as RTE and Clear. The composite analysis provided the confirmatory test that RTE days are the days with visible aerosol transport from the Indo-Gangetic Plain (IGP) towards the southern Indian peninsula, as depicted in Supplementary Figure S1, are included in the analysis. While, the clear days are the days without long-range transport from North-India. The variability in MODIS-observed daily AOD values within the east coast box was analysed and it was found that AOD values equal to 0.7 and 0.3 represent the 70th and 30th percentile values, respectively**

**Moreover, it is important to note that days affected by cloud cover are excluded (i.e days with mean Cloud Fraction > 0.1) from the segregation process to ensure the reliability of both RTE and clear day classifications.**

Comment: The methodology section is weak and lacks details. I recommend to rewrite this section and provide clear and elaborate explanations for all methods, including RTE categorization, endurance estimation, and aerosol transport identification.

**Reply: We thank the reviewer for the suggestion to improve the methodology section. We have rewritten the section as suggested to provide detailed explanations of RTE categorization and aerosol transport identification in the revised manuscript. The HYSPLIT trajectory analysis and prevalent wind system shows evident haze transport from the northern parts of the India.**

**As per reviewer's suggestion, the endurance estimation of RTE events is also elaborated, and provided in detail to the comment followed.**

Comment: On what basis is the endurance of RTE days estimated? Specify the data source used to determine the endurance period. What criteria or threshold are used to distinguish clear days from RTE days on a diurnal scale? Are there any previous studies reporting RTE endurance over the Indian domain? Citing prior works could strengthen your findings.

**Reply: The endurance of RTE days is estimated based on the prevalence of mean AOD obtained from MODIS observation, exceeding 0.7 over the eastern-coastal box, as shown in Fig.1. Figure 1a shows the composite of the RTE days estimated between 2015 and 2024. The RTE characterization are based on a day, as MODIS AOD products (Terra+ Aqua combined) are available for 1 day period only; hence diurnal variation of such aspects are not attempted using MODIS.**

As mentioned in the manuscript, the RTE generally occurs for 1 day to 4-6 days. However, we observed a prolonged RTE event sustained for 12 days during 2022. Supplementary figure Fig.S2a shows the year wise number of observations of RTE episodes during the winter season lapsed for 1 day to 5 days and, for more than 5 days. Our motivation was to highlight the endurance period of the RTE events and hence we weighted the number of observations with the duration day bins (N multiplied by day bins) as shown in Fig.S2b. For instance, the total observation of RTE endurance during 2015 for 1 day to 4 day are equal (Fig.S2a), but Fig.S2b signifies the duration of such episodes by adding weights to the day bins. Figure S2c shows the similar weighted observation as shown in Fig.S2b; however, shown for 1 day, and cumulative weighted observation between 2-4 day and more than 4 days. The discussions are included in the revised manuscript as a supplement.

[Figure]

*Figure S2. Annual variation of RTE occurrence duration. (a) Total number of observations (N) obtained for 1 day, 2 day, 3 day, 4 day, 5 day and more than 5 day (colour coded separately). (b) The number of observations (N) weighted by the day bins, shown separately for different day bins. (c) Same as (b) but shown the cumulative counts for 1 day (black), 2-4 day (magenta) and more than 4 day (red)*

We agree with the Reviewer that, there might be diurnal variation as these RTEs is controlled by the stable boundary layer. However, we also see that the RTE episodes are generally phenomena

**spanning from 1 to 5 days, so the endurance is calculated at daily resolution. This study provides a first observational framework for the transported pollutants and its impact on the boundary layer dynamics over Chennai; we have not come across with the similar observational evidence of the transport, as of our knowledge.**

Comment: The region referred to as the "south-eastern coast box" is not shown in any figure. Include this region clearly in a figure for better understanding.

**Reply: Thanks for the comment. The region is added in the revised manuscript as shown in Figure 1 of the manuscript.**

Comment: The RH on clear days is relatively higher than on RTE days. How does hygroscopic growth contribute to the increase in endurance during RTE periods, given these RH differences?

**Reply: We thank the reviewer for this thoughtful comment. However, the RH differences during RTE and clear days alone are not sufficient to infer the endurance of the RTE period in regard to the hygroscopic growth of the aerosols. Given the limited samples and instrumentations to address such factors, the statements relating to the hygroscopic growth and influence of RH on the RTE and TAL are removed from the revised manuscript.**

Comment: Differences in wind fields between RTE and clear days alone may not be sufficient evidence for aerosol optical depth (AOD) enhancement over the southeastern coast and peninsular region. Provide additional supporting evidence or analysis to strengthen this.

**Reply: We agree to the reviewer that the difference in the wind field during the RTE and clear day are not only factors for the AOD enhancement over the eastern coast, rather they also influenced by the aerosol loading over northern India and inland, also its transport towards the southern Indian peninsula. Figure 1e and f depicts the number density of backward trajectories obtained from the HYSPLIT back trajectory model and it shows that the aerosols observed over the southern Indian peninsula are predominantly transported from northern India during the RTE days compared to the clear days; however, it limits to the nearby oceanic region during the clear days.**

**Further, we also examined the change in the Aerosol Direct Radiative Forcing (ADRF) during the RTE and clear days, where aerosols effectively respond to the radiation, and provided in supplementary figure Fig.S6. Figure S6 shows the ADRF observed during the RTE and clear days, estimated from MERRA2 radiative flux observations at the surface (Thomas et al, 2019;2021). Overall, the ADRF has a net cooling at the surface both during the RTE and clear days. However, RTE episodes triggers to enhance the cooling of the surface to ~-20-40 W/m². Such strong cooling is observes to be around the eastern coastal regions where the aerosol transports generally occur. In specific, it reduced to less than -40 W/m² over the eastern coastal box where the TAL present.**

During clear days (Fig.S6b) the strong cooling is confined over the IGP alone (~-25-30 W/m²). The difference in the ADRF during the RTE and clear day composite is shown in Fig.S6c, evidencing a cooling of ~-15-20 W/m² by the RTE days. It also suggests that aerosol transport from the north India has a profound effect on the radiation and eventually on the ABL development and hence in the PM distributions.

[Figure]

*Figure S6. Aerosol Direct Radiative Forcing observed during the(a) RTE and (b) Clear days. (c) Difference between the ADRF of RTE and clear days*

Comment: Line 205: Define TAL (Transported Aerosol Layer) when it first appears in the text.

**Reply: As suggested, TAL is defined on its first appearance in the revised manuscript**

Comment: Lines 241–257 Why were total mean diurnal changes studied instead of individual day differences? Analyzing day-specific differences may offer greater insights into variability.

**Reply: We thank the reviewer for insightful suggestion. As suggested, the individual day difference along with the diurnal changes in the year 2018 alone (to avoid the interannual variations) is added in the revised manuscript and shown in Fig. 2d&e respectively.**

**Figure 2d shows the day averaged profiles of RTE (24,5,26,27 Jan 2018) and clear days (23,28,29, Jan 2018) observed during the typical case from MPL observations, as shown in Fig.2c. Although the extinction values is observed to similar near the surface, it rapidly decreases till ~0.8 km during the RTE days. Further, it maximizes within the altitude range ~1-2.5 km. Overall, the aerosol extinction during the RTE days are observed to be 50-60% higher during the RTE days than clear days, maximising between 1-2.5 km. Such enhancement can be attributed to the presence of TAL. The temporal changes the difference of extinction coefficients between RTE and clear days during the year 2018 is shown in Fig.2e. Similar to the typical events, the extinction coefficient during the RTE days are observed to be more than 0.25 during the forenoon hours near the surface. However, the enhancement observed above the ABL consistently maintains throughout the day.**

[Figure]

*Figure 2(c) Time-Altitude cross-section of the total attenuated extinction coefficient obtained from Micro Pulse Lidar (MPL) observation over Chennai (SRM IST) between 23 and 29 January 2018. The black line corresponds to the temperature profiles from radiosonde over IMD, Chennai. The black dotted line corresponds to the derived ABL-H, and the red dotted lines are the top of the TAL. (d) Mean extinction coefficient observed during a typical RTE (24,25,26,27 Jan 2018) and clear day (23,28,29 Jan 2018) estimated from MPL observation. The difference between the extinction coefficient observed during RTE and clear day is shown as a dotted line (axis on the top). (e) Temporal changes in the difference between the extinction coefficients during the RTE and clear day composite in 2018.*

Comment: Care should be taken to reduce the several typo errors in the manuscript.

**Reply: The revised manuscript is thoroughly checked for any typos and corrected wherever it is.**

Comment: Figure 1: Panels (a, b): Why is there a data gap in the bottom-left corner (65°E–70°E and 5°N–10°N)? Panels (c, d): At what altitude are the wind vectors shown? Why are wind vectors missing over the Tibetan Plateau during RTE days, while they are visible on clear days? Panel (e): The description is unclear. How are the weighted counts estimated? Are they based on monthly data or for the entire study period? Panel (f): Why are wind speed differences over high-altitude regions masked?

Justify this masking and its impact on interpretation. Suggest adding Chennai and Karaikal locations on Figure 1 for geographic reference.

**Reply: We checked it thoroughly and corrected them in the revised manuscript. The pointwise responses are provided as follows:**

*Panels (a, b): Why is there a data gap in the bottom-left corner (65°E–70°E and 5°N–10°N)?*

**This data gap in the spatial distribution of MODIS AOD product is due to the unavailability of datasets in the provided grid box. This region is primarily ignored because it does not come in our area of interest. The number of datasets available in the 0.5-degree grid box (lon x lat) for both the RTE and clear days are provided in Fig R1. We also would like to mention that this area is not masked intentionally.**

[Figure]

***Figure R1. Availability of data samples of MODIS AOD product in a 0.5-degree grid box during both RTE and clear days between 2015 and 2024.***

*Panels (c, d): At what altitude are the wind vectors shown? Why are wind vectors missing over the Tibetan Plateau during RTE days, while they are visible on clear days?*

**The wind vectors are shown at 850 hPa, as mentioned in the figure description. The mean wind vectors of available RTE and clear days are shown in panels c and d respectively. The wind vectors across the Tibetan Plateau (TP) are available only for 4 days, hence it is eventually averaged while generating the figures. We revised the figure Fig.1b and c by removing such days since the TP does not have data for at least 70% of the observation times**

[Figure]

*Figure 1. The total aerosol extinction (TAE) from MERRA2 reanalysis dataset observed for the composite of (b) RTE and (c) clear days superimposed with the wind vectors during the respective events*

*Panel (e): The description is unclear. How are the weighted counts estimated? Are they based on monthly data or for the entire study period?*

**The RTE episodes are categorized based on the exceedance of mean AOD within the eastern coastal box and it may last for a day to more than 12 days between December to March. Here we attempted to capture the endurance of such events based on the MODIS observations, irrespective of the months. Some events may start to be observed end of a month and last till the first week of the preceding month. Hence, the classification of the endurance of such events is per year.**

**We tracked the endurance of RTE events, taking the initial occurrence as day 1, till the eastern coastal box became clear day. There are multiple times an RTE occurs in an year, and their duration can be for 1 day up to 12 days. These multiple occurrences are termed as counts (N). The weighted counts are estimated by multiplying the N by the respective days. For instance, the total observation of RTE endurance during 2015 for 1 day to 4 day are equal (Fig.S2a), but Fig.S2b signifies the duration of such episodes by adding weights to the day bins. Figure S2c shows the similar weighted observation as shown in Fig.S2b; however, shown for 1 day, and cumulative weighted observation between 2-4 day and more than 4 days. The following discussions are added in the revised manuscript as a supplement.**

**Figure 2:** Is it possible to highlight the region of interest on the map where the CALIPSO profiles are considered? This would improve clarity. Were there any close overpasses of CALIPSO to the MPL station? If yes, how consistent are the CALIPSO profiles when compared to the MPL observations? Clarify how the Atmospheric Boundary Layer Height (ABL-H) and Transported Aerosol Layer (TAL) are identified from MPL profiles.

**The region of interest of CALIPSO passes are highlighted in Figure 1 in the revised manuscript, as the eastern coastal box. The vertical distribution of aerosols obtained from MPL and CALIPSO**

are validated using the profiles having closest proximity to the MPL and presented in our earlier publications (Ananthavel et al., 2021a, 2021b).

The diurnal variability of ABL-H is estimated using the Wavelet Covariance Transformation (WCT) method (Reddy et al., 2021; Davis et al., 2000; Pal et al., 2010; Baars et al., 2008), which estimates the ABL-H from lidar profiles by step changes in signals using Haar function. The TAL is identified using the differential zero crossing method (Mehta et al., 2023; Ali et al., 2022), similar to the methodology followed by Mehta et al., 2023 to identify the elevated aerosol layer. In general, the extinction coefficient gradually decrease above the ABL. However, presence of TAL can increase the extinction values similar to as observed within the ABL. The differential zero crossing method identifies the top of the TAL using the gradient of extinction coefficient profiles. Note that, this method of TAL detection is used only when a valid ABL is identified.

**Figure 4:** Panel (a): Is this plot representative of Chennai or Karaikal? Provide clarification. Panel (c): Is the exponential fit used in the figure the most appropriate fit for the data? If possible, justify the choice of the exponential model or test alternative fits for robustness. Provide clarity regarding climatology values of ABL-H and PM2.5. Specify the data sources for these values. Indicate which data points in the figure correspond to RTE and clear days to improve interpretability.

The plot is representative of Chennai alone. The analysis is not extended over Karaikal due to the non-availability of datasets such as surface weather observations, PM and MPL.

[Figure]

*Figure R2. Scatter of PM 2.5 anomaly (%) and ABL-H anomaly over Chennai with the linear and exponential fits.*

Figure R2 shows the scatter and both the linear and exponential fit of the relation between PM2.5 and ABLH anomaly. Since the r-squared value of the exponential model fit is 0.25 compared to the linear fit (0.18), the exponential fit is used to infer the relation between PM 2.5 and ABL-H anomaly over Chennai. Such approaches are followed by previous researchers for similar dust transport events in China (Su et al., 2020).

The ABL-H as shown in the panel b is derived from radiosonde thermodynamic profiles (Mehta et al., 2017), whereas panel c is solely derived from MPL observations over SRM IST (Reddy et al., 2021). PM 2.5 is obtained from US Consulate, Chennai through the AirNow data archive. Please note that the presented relation between PM2.5 and ABLH in panel c is shown only for the cases of RTE (N=255).

As discussed earlier, the term climatological values of ABL-H and PM 2.5 are inappropriate to use and hence it is changed as "mean ABL-H between 2016 and 2019" for the ABL-H, and "mean PM 2.5 between 2015 and 2024" for the PM 2.5. While mean ABL-H is estimated from MPL observations, PM 2.5 is over US Consulate, Chennai through AirNow data archive. The following discussions are added in the revised manuscript with additional information from the various radiosonde observations across the eastern coast.

Comment: Provide additional figures and clarifications, such as: Highlighting CALIPSO regions of interest in Figure 2. Marking the south-eastern coast box explicitly. Comparing CALIPSO overpasses to MPL observations.

The manuscript is thoroughly checked and revised as per the constructive comments provided by the reviewer. A comparison between the CALIPSO and MPL during an RTE day is provided in Figure R3.

[Figure]

*Figure R3. (a) Mean AOD obtained from MODIS between 24-27 January 2018 - typical RTE days. Wind vectors are superimposed over the figure and the magenta line corresponds to the CALIPSO overpass observed on 25-01-2018 (b) Latitude- altitude cross-section of the total attenuated backscatter obtained from CALIPSO on 25-01-2018 (nighttime). The vertical magenta corresponds to the location of SRM IST, where MPL is installed. (c) Vertical variation of aerosol extinction coefficient obtained from MPL on 25-01-2018 between 0-6 IST. The vertical magenta line corresponds to the time of pass of CALIPSO near the station.*

Figure R3a shows the spatial distribution of MODIS-AOD during RTE days − 24-27 January 2018, showing a substantial increase in the AOD over the eastern coast, attributed to the aerosol transport from northern India towards the inland regions of the southern Indian

peninsula. The magenta line corresponds to the CALIPSO overpass observed on 25-01-2018 during its descending mode (nighttime). Fig.R3b shows the latitude-altitude cross-section of the total attenuated backscatter between 12-14 degree latitude band- nearest proximity of SRM IST (at ~01:40 IST) where the MPL is situated. It can be directly observed an accumulation of aerosols within 0.5 km, attributed to confinement of the aerosol within the boundary layer. The aerosol observed till ~2.3 km, attributed to the presence of TAL. Fig.R3c shows the aerosol extinction coefficient observed over Chennai (SRM IST). Similar to the CALPSO attenuated backscatter observation, MPL also shows the maximum confinement of aerosol within ~0.5 km as the ABL (shown as the black dotted line) observes at the altitude. The observed TAL, as a layer hangs above the ABL till ~2.3 km, is also observes similar to the CALIPSO detected TAL.

**References:**

Thomas, A., Sarangi, C., & Kanawade, V. P. (2019). Recent Increase in Winter Hazy Days over Central India and the Arabian Sea. *Scientific Reports*, *9*(1), 1-10. https://doi.org/10.1038/s41598-019-53630-3

Thomas, A., Kanawade, V. P., Sarangi, C., & Srivastava, A. K. (2021). Effect of COVID-19 shutdown on aerosol direct radiative forcing over the Indo-Gangetic Plain outflow region of the Bay of Bengal. *Science of The Total Environment*, *782*, 146918. https://doi.org/10.1016/j.scitotenv.2021.146918

---

## Author Comment (AC2)

**Overview/General Comment**

The manuscript entitled "Regional transport of aerosols from Northern India and its impact on boundary layer dynamics and air quality over Chennai, a coastal megacity in Southern India." by Ali et al. attempted to study the critical influence of aerosols on boundary layer dynamics. The research problem is highly significant and holds considerable importance within the scientific community. However, the results are very loosely presented with inadequate justification. The present analysis ranging from ABL identification to TAL transport, relevant interpretation requires greater depth and clarity to attempt the present problem. The estimation of ABL-H, ABL-AOD need to be revised in all the sections before presenting the analysis. Given these shortcomings, I believe substantial revisions are necessary. Therefore, I recommend resubmission, provided the authors address the following comments,

**Reply: We thank the reviewer for critically evaluating our manuscript and providing constructive comments. We have carefully gone through the suggestions and incorporated changes in the revised manuscript. As suggested, description of the ABL-H and ABL AOD are revised.**

**Specific Comments**

Comment: Line no 105: The statement "during December – March between 2015 and 2024" is inaccurate as the CALIPSO mission ended on August 1, 2023. needs clarification on this statement.

**Reply: We thank the Reviewer for pointing it out the typo. It is corrected as "… December - March between 2015 and 2023" in the revised version.**

Comment: Line no 153: The authors refer to "The climatological mean spatial distribution of columnar AOD." However, defining 2015 to 2024 as a climatological period is inappropriate. Please revise accordingly.

**Reply: The term "climatological mean" is replaced by "10 year mean"**

Comment: Line no 158-160: The statement, "The RTE and clear days are observed to be 119 and 70 days... RTE shows an enhancement of more than 0.7 over the eastern coast of India compared to clear days," requires clarification: (1) What latitude/longitude range was averaged to derive the value of 0.7? (2) Is the RTE enhancement consistently above 0.7 for all 119 days?

**Reply: (1) We have calculated the RTE and the clear day over the east coast region mostly affected by the haze which is marked as a rectangular box as depicted in Fig.1b of the revised manuscript. The RTE days are the days during the months of December to March, for which domain-averaged AOD within this rectangular box in the east coast exceeds 0.7. Similarly, the days when the domain-averaged AOD is less than 0.3 is taken as clear days. Note that background AOD remains higher than 0.3 during December to March. So, we define it as clear days to differentiate it from the RTE days. The variability in MODIS-observed daily AOD values within the east coast box**

was analysed and it was found that AOD values equal to 0.7 and 0.3 represent the 70[th] and 30[th] percentile values, respectively. In addition, the spatial distribution of aerosols during RTE days are also inspected manually to confirm that pollution transport from North India using back trajectory analysis. Accordingly, 119 and 70 cases of RTE and clear days are obtained during 2015-2024.

(2) Yes, the domain-averaged AOD within the rectangular box (eastern coastal box) is consistently above 0.7 for the RTE days observed.

[Figure]

*Figure 1 Composites of AOD observations from MODIS shown separately for (a) RTE and (b) Clear days. The region to identify the RTE and Clear days is     shown as a dotted box, named as an eastern coastal box.*

Comment: Line no 195-197: "decreasing gradient in aerosol extinction values between surface and 2 km" between RTE and Clear days. This observation is very important. However, it is also important the uncertainty associated with it, (i) author mentioned Fig 2a and 2b is a mean extinction between Dec and March 2015 to 2023 and the (ii) passes considered for the present study is ± 5 deg longitudes which include land and ocean.  Considering that the following uncertainty arises, December features strong ABL inversion (~1 km), while March inversions can extend to ~2.5 km. Similarly, oceanic passes may yield aerosol peaks near 1 km, whereas land passes could show concentrations up to 2 km. I recommend presenting RTE and clear-day analyses for individual months and separating land and ocean data to address these uncertainties.

Reply: We thank the reviewer for the valuable constructive comment for an in depth analysis of the land-sea contrast on aerosols gradient concerning ABL variation. To demonstrate clarity on the understanding, we have estimated the mean extinction coefficient from CALIOP for the RTE days and the clear days over land and ocean separately and have depicted/discussed it now as supplementary figures in the revised manuscript (discussed below). Further, we also attempted to do the same for monthly time scale during December to March.  However, the CALIOP are

highly sparse at monthly scale so we could not find adequate understanding on the temporal changes in the aerosol gradient in a monthly scale

Supplementary figure Fig.S7 and S8 show the spatial distribution of aerosol across the eastern coastal box during the RTE and clear days, respectively. It also shows the number of valid aerosol data samples available are ~10-15 for the entire period December and March 2015 to 2023, to obtain the mean extinction coefficient over the box. The composite domain-mean and column averaged aerosol extinction coefficient over land and sea was observed to be ~0.45 and 0.2, respectively, during the RTE days. In comparison, the composite mean aerosol extinction coefficient over land and sea during clear day was observed to be ~0.25 and 0.1, respectively. However, it observes more than 0.45 over the IGP regions during both the RTE and clear days. It is also interesting to observe that the extinction gradient is higher near the coast compared to away from it.

Fig. S7c shows the vertical variation of the aerosol extinction indicating the higher aerosol concentration below the 2 km which features the stronger concentrations with increasing latitude in the selected box, where the strong aerosol accumulation rises till 2.5-3.5 km. However, the aerosol are mostly concentrated below 1 km during clear days (Fig.S8c), confined to the higher latitudes only. Fig.S7e suggests that the overall gradient present in the extinction coefficient is mainly contributed by the land part. Since the marine boundary layer is often formed below 500 m, CALIOP observations are limited and have large bias (Ananthavel et al., 2021) to fully resolve the distinction in the gradient over the ocean. However, the aerosol extinctions over ocean during the RTE days (Fig.S7f) are observed to be higher (~0.2) than clear days (Fig.S8f). It can also be noted an enhancement in the aerosol extinction coefficient at elevated heights (~1-3 km). Such enhancements can be attributed to the aerosol loading due to the transport. On the other hand, enhancement in the aerosol extinction coefficients are due to the aerosol loading within the boundary layer (<1 km) and also due to the TAL presence (~1-3 km)

The differences in the aerosol extinction coefficient between RTE and clear day composites are shown in Fig. S9, separately shown for the land and aerosol composites. It also suggests that the TAL enhances overall aerosol loading over the eastern coast, contributing to an enhancement of 0.2-0.3 compared to clear days, mostly confined over the land.. On the other hand, it observes above the ABL (>1.5 km ) over the oceanic region.

[Figure]

*Figure S7. Spatial distribution of CALIOP-observed aerosol extinction coefficient along the eastern coast during the RTE days. (a) Number of valid aerosol data samples available in 1 degree x 0.1 degree (lon x lat) grid box. (b) mean extinction coefficient and (c) vertical variation of aerosol extinction coefficient within the eastern coastal box portrayed in (a) and (b) for the RTE composites. (d) the land and ocean separation within the eastern coastal box. Latitude-altitude cross-section of the aerosol extinction coefficient over (e) land and (f) ocean.*

[Figure]

*Figure S8. Same as Fig.S7 but for clear days*

[Figure]

*Figure S9. Difference in the extinction coefficient between RTE and clear days obtained separately for (a) Land and Ocean composites (b) Land composites alone and (c) ocean composites alone*

Comment: Line no 210: "Can be seen above the ABL during the RTE periods" I recommend to show the 3 or 5 days backward trajectory analysis from surface upto 4 km during RTE and Clear sky days.

**Reply: We thank the reviewer for the suggestion to show the back trajectory analysis for RTE days. We have used the National Oceanic and Atmospheric Administration (NOAA) Hybrid Single-Particle Lagrangian Integrated Trajectory (HYSPLIT) model to compute the 5-day backward trajectories to understand the origin of the TAL over Chennai. Figure 1e-f shows the number density of 5-day backward trajectories within 4 km from the surface, computed for every 6 hours of a day during RTE and clear days over Chennai. It suggests that the aerosols observed over Chennai during RTE days originated primarily from the northern Bay of Bengal and the land areas across the eastern coast. On the other hand, the aerosols over Chennai are mostly transported from the nearby regions of the Bay of Bengal on clear days.**

[Figure]

*Figure 1. Number density of 5-day backward trajectories initiated from Chennai between surface and 4 km during (a) RTE and (b) Clear days.*

Comment: Line no 213 – 214: "temporal variation ….AOD" I have noticed several instance the ABL AOD towards Zero. This reflects the ABL estimation in figure 2c has been associated with uncertainty. For instance, on 27-01-2024 after 18 LT the ABL height and the first altitude bin are at the same altitude. In real atmosphere it never exists. This needs to be taken care in all the section of ABL height identification otherwise the claim of ABL height reduction, boundary layer dynamics may not be valid

. It is also to be noted the entire manuscript depends on ABL identification to discuss the TAL and BL dynamics.

**Reply: We thank the reviewer for pointing it out. We have re-calculated the ABL height and modified the Figure 2 in the revised manuscript. We acknowledge that such confusions are raised due to the bigger size of line used to depict the ABL-H. Numerous instances the ABL height during nighttime becomes very low, and sometimes even less than 300 m which causes difficulty in attributing the accurate ABL height. For such cases, we imposed continuity conditions and the average ABL height detectable just before and after it. The manuscript is revised with correcting the line size of ABL-H and starting the altitude index from the surface (above ground level) as shown below. In addition, here we would like to clarify the methodology adopted for ABL and AOD estimation.**

**The MPL is installed on top of a building with a height of 80 m. Additionally, the lidar has a blind zone up to 230 m due to the saturation of photomultiplier tubes and high reflection from near-field objects. Consequently, the MPL is effectively blind to the surface up to ~0.3 km in the real atmosphere. Therefore, the MPL backscatter data presented here do not correspond to the first altitude bins. Despite this limitation, the high vertical resolution of the MPL (30 m) enables the detection of minute variations in the aerosol vertical distribution.**

[Figure]

*Figure 2c. Temporal variation of extinction coefficients estimated from MPL observations between 23-29 January 2018*

**For the identification of ABL height (ABL-H), we adopted the Wavelet Covariance Transformation (WCT) method, a widely used approach for ABL detection from lidars (Reddy et al., 2021; Mehta et al., 2023; Davis et al., 2000; Pal et al., 2010; Baars et al., 2008). This method is effective in detecting the Stable Boundary Layer (SBL), Convective Boundary Layer (CBL), and Residual Layer (RL). The potential of the WCT method and MPL for detecting the ABL over Chennai has been demonstrated in our earlier publications (Mehta et al., 2023; Aravindhavel et al., 2021a, 2021b; Reddy et al., 2021). However, we rechecked the WCT method thoroughly and confirmed that first altitude bins are not taken as the ABL-H.**

**The ABL-AOD are observed to be quite low due to very shallow ABL-H, especially when a denser TAL present above it. Also, the low values of aerosol extinction under very shallow ABL-H can eventually reduce the ABL-AOD. To avoid such confusion, we now analyse AOD above the ABL (ABL$_{FT}$) instead of below the ABLH in the revised manuscript. The supplementary figures showing the ABL-AOD also corrected accordingly and shown below.**

[Figure]

**Figure S4a. Temporal variation of columnar AOD and AOD above the ABL between 23-29 Jan 2018**

Comment: Line no 217: "..ABL (integrated extinction within surface and ABL-H) decreases from ~0.4 to less than 0.2.." re-estimation of ABL altitude, ABL-AOD needed.

**Reply: The ABL estimation is based on the Wavelet Covariance Transformation (WCT) method, a widely adopted methodology for lidar-based ABL detection and well documented in our earlier publications (Reddy et al., 2021; Mehta et al., 2023; Davis et al., 2000; Pal et al., 2010; Baars et al., 2008). It is to be noted that the MPL is incapable for providing backscatter information below 0.3 km due to its blind zone. In reality, the Stable Boundary Layer (SBL) or Convective Boundary Layer (CBL) can appear at approximately 0.3 km during their developing stage, particularly in the early morning hours (Mehta et al., 2017; Mehta et al., 2023). However, as replied to the earlier comment, we ensured that first altitude bins are never counted as ABL-H. Since ABL-AOD estimation depends on the ABL height (ABL-H), it tends toward zero when the ABL-H is close to 0.3 km. It generally occurs when a deep TAL occurs above the ABL. The dramatic decrease of ABL-AOD, during such instances, are meaningless and hence the term ABL-AOD is replaced by the AOD above ABL, in the revised manuscript. The appropriate changes have been made throughout the revised manuscript.**

Comment: Line no 220-221: "….the altitude, where the TAL presents, is observed to be warming…." "…contrary to the earlier findings, the surface temperature is also observed to be warming …" it is confusing. 1) What analysis supports the warming conclusion? Temperature profiles typically decrease with height in Fig 2c. 2) Which earlier findings author mentioning? As similar how authors concluding the surface temperature warming? Is author mentioning the diurnal variation ? with respect to what it is warming ?

**Reply: This statement is based on Fig.3, panel c and f in the manuscript, where we discussed the differences in the vertical temperature during RTE and clear days. We observed that the entire column from the surface to top of TAL is warmer during the RTE days compared to clear days. It is observed that the temperature between ~0.5- 2 km over Chennai and Karaikal is warming and during RTE days. The relative warming compared to clear days is up to ~2.5 K.**

**The earlier findings are referred to the aerosol dome effect (Ma et al., 2022, Ma et al., 2020). Aerosol layer aloft the ABL, in general, absorb solar radiation and heats the atmosphere at the altitude where it present. Eventually, it declines the radiation reaching to the surface inducing surface cool (Sarangi et al., 2016; Ma et al., 2022). However, please note that here, instead of surface temperature, radiosondes are measuring air temperature. Previous studies have shown that the aerosol induced stability effect impacts the convective dissipation of heat from boundary layer to free troposphere, causing a warmer air temperature in boundary layer during high aerosol loading days (Feng et al., 2016). In accordance, we observe the surface temperature to be warming during the RTE days (as shown in supplementary figure Fig.S4 in revised manuscript) compared to clear days.**

Comment: Line no 224: "..ABL-H also decreased from ~1.4 km to ~0.3 km.." I can see the ABL temperature inversion of at ~ 1.0 km during 25-01-2018 from the radiosonde observations.

**Reply: We apologize for the mistake appeared while plotting the temperature and MPL-extinction profiles together. Correction applied to the altitude bins for both the MPL and temperature profiles and showed in Fig.2c in revised manuscript.**

Comment: Line 225: "The climatological ABL-H".. can be replaced to "The mean ABL-H.." use of climatological may not valid.

**Reply: Thanks for notifying this error. The term "climatological ABL-H" is corrected as "mean ABL-H" everywhere.**

Comment: Line no 226-227: "RTE events in altering the boundary layer dynamics" it is loosely stated, I recommended authors should propose the mechanism, appropriate explanation with proper evidence.

**Reply: Although the association of the RTE events in altering the boundary layer is established in the section 3.2 and 3.3, we find the usage of this statement is inappropriate here. Hence the statement "indicating the role of RTE events in altering the boundary layer dynamics" is removed.**

Comment: Line no 237: "strong inversion at the altitude of ABL and the top of the TAL during the hazy days, attributed to the aerosol heating" the strong inversion layer is related to the large-scale seasonal variation Sinha et al (2013), Ganguly et al (2006). Interpretation needs to be revisited.

**Reply: We agree with the reviewer about the confusion in this sentence due to the phrase "altitude of ABL and the top of the TAL". Hence, we revised the sentence for clarity as, "The temperature profiles (Fig.2c) show a strong inversion at the top of the TAL can be attributed to the aerosol-induced warming effects. The strong inversion layer formation can also be related to the large-scale seasonal variations (Sinha et al., 2013; Ganguly et al., 2006). However, such aspects are not addressed here due to the limited datasets. The observation suggests that the presence of strong inversion above the ABL can also be attributed to the presence of TAL, followed by widespread haziness and pollution."**

Comment: Line no 242-243: "RTE (10 days) and clear (6 days) day cases within 2018 and 2023 obtained from MPL". I recommend the author to present within a month or within the season of particular year to avoid the uncertainty related to inter and intra variability.

**Reply: We agree to the reviewer and the Fig.2d in the manuscript is revised by presenting the observation within the year 2018 alone, where the total number of occurrences of RTE and clear days are high.**

[Figure]

*Figure 2*(c) Time-Altitude cross-section of the total attenuated extinction coefficient obtained from Micro Pulse Lidar (MPL) observation over Chennai (SRM IST) between 23 and 29 January 2018. The black line corresponds to the temperature profiles from radiosonde over IMD, Chennai. The black dotted line corresponds to the derived ABL-H, and the red dotted lines are

**the top of the TAL. (d) Mean extinction coefficient observed during a typical RTE (24,25,26,27 Jan 2018) and clear day (23,28,29 Jan 2018) estimated from MPL observation. The difference between the extinction coefficient observed during RTE and clear day is shown as a dotted line (axis on the top). (e) Temporal changes in the difference between the extinction coefficients during the RTE and clear day composite in 2018.**

**Figure 2d shows the day averaged profiles of RTE (24-27 Jan 2018) and clear days (23,28,29, Jan 2018) observed during the typical case from MPL observations, as shown in Fig.2c. Although the extinction values is observed to similar near the surface, it rapidly decreases till ~0.8 km during the RTE days. Further, it maximizes within the altitude range ~1-2.5 km. Overall, the aerosol extinction during the RTE days are observed to be 50-60% higher during the RTE days than clear days, maximising between 1-2.5 km. Such enhancement can be attributed to the presence of TAL. The temporal changes the difference of extinction coefficients between RTE and clear days during the year 2018 is shown in Fig.2e. Similar to the typical events, the extinction coefficient during the RTE days are observed to be more than 0.25 during the forenoon hours near the surface. However, the enhancement observed above the ABL consistently maintains throughout the day.**

Comment: Line 249-255. "….significantly reduce the short-wave radiation of incoming solar radiation, hence reducing the surface heat flux and development of ABL…." The author's claim is not relevant from the present observation. I recommend perform flux direct observations or at least from reanalysis data to support this statement.

**Reply: We thank reviewer for this comment. Figure S6 shows the aerosol direct radiative forcing (ADRF) observed during the RTE and clear days, estimated from MERRA2 radiative flux observations at the surface (Thomas et al, 2019;2021). Overall, the ADRF has a net cooling at the surface both during the RTE and clear days. However, RTE triggers to enhance the cooling of the surface to ~-20-40 W/m². Such strong cooling is observes to be around the eastern coastal regions where the aerosol transports generally occur. In specific, it reduced to less than -40 W/m² over the eastern coastal box where the TAL present. During clear days (Fig.S6b, the strong cooling is confined over the IGP alone (~-25-30 W/m²). The difference in the ADRF during the RTE and clear day composite is shown in Fig.S6c, evidencing a cooling of ~-15-20 W/m² by the RTE days. It also suggest that aerosol transport from the north India has a profound effect on the radiation and eventually on the ABL development and also in the PM distributions.**

[Figure]

*Figure S6. Aerosol Direct Radiative Forcing observed during the(a) RTE and (b) Clear days. (c) Difference between the ADRF of RTE and clear days*

Comment: Line 259 – 298: Section 3.2. I did not understand why the author decided to present the meteorological observation between Chennai and Karaikal. When the TAL is from the IGP?. Spatial analysis, or data from stations along the transport pathway from IGP to Chennai, would be more appropriate.

**Reply: We thank reviewer for this constructive comment and the analysis is extended over the eastern coast rather limiting to the Chennai and Karaikal. We have now revised the manuscript and included radiosonde analysis from 5 IMD stations along the eastern coast as shown in revised Figure 3 (shown below) which captures the North-South gradient of the TAL transport from IGP to South India. The new analysis provides clear enhancement in the air temperature (~ 0.5-2 K) at elevated altitudes (0.5 - 3 km) where the TAL is present during the RTE (w.r.t. the results seen in Figure 2).**

**Over Kolkata and Bhubaneswar, where it lies in the northmost region, the relative difference in the temperature RTE ($\frac{T_{RTE}-T_{Clear}}{T_{RTE}} \times 100\%$) maximizes around 1.5 km and sharply decreases afterwards. Over Vizag, such decline occurs above 2.5 km. Interestingly, Chennai and Karaikal, the south most stations, observes the heating throughout the column up to 3 km. Over Chennai, the altitude of observed peak warming coincides with the peak occurrence of TAL, as observed in Fig.2, suggesting the aerosol-induced radiative effects of TAL on temperature profiles. This phenomenon also suggests that aerosol-induced warming at the lower troposphere not only enhance the temperature at the altitude where aerosol occur but also modifies the overall temperature profiles of the lower atmosphere. Moreover, the observed phenomena of enhancement in altitude of RTE-associated warming as we move southward from IGP, is also consistent with the fact that the long-range transported aerosol plumes gets elevated at downwind locations (Yu et al., 2012; Stohl et al., 2006).**

**We also observed heterogeneity in the magnitudes of RTE-associated warming across these stations. Greater warming (2- 2.5 K) was observed at Kolkata, Chennai and Bhubaneswar compared to Vizag and Karaikal, indicating that the RTE-impact on warming (on RTE days relative to clear days) is greater for megacities.**

[Figure]

*Figure 3. Vertical profiles of temperature during RTE (red) and Clear (blue) days obtained over the different station along the eastern coast. (a) Locations of the Radiosonde observations. (b)-(f) mean temperature profiles with standard errors over the stations Kolkata, Bubaneswar, Vizag, Chennai and Karaikal during RTE and Clear days, and the relative difference between the RTE and Clear in percent (axis on the top). Vertical dashed line corresponds to the 0%.*

Comment: Section 3.3. The analysis presented is insufficient to substantiate discussions on BL dynamics. Consider including spatial analyses from CALIPSO or other reanalysis datasets to strengthen this section.

**Reply: We thank reviewer for this constructive comment. This work is aimed to provide observational evidence of transported aerosols on the boundary layer dynamics and consequential air quality degradation over megacities. As reviewer suggested, CALIPSO provides a spatial distribution of aerosols; however, it is limited to ~01:30 AM/PM, hence, CALIPSO along**

with reanalysis data cannot be used due to limited data. However, we have included additional analysis in the revised manuscript using radiosonde measurements from 5 different IMD sites along the east coast to show the signatures of RTE-associated changes in boundary layer height compared to background clear conditions

The warming observed during the RTE is expected to enhance the vertical temperature stratification thus impeding the boundary layer height, commonly termed as "dome effect" (Ding et al., 2016). Figure 4a shows the probability distribution of lower tropospheric stability (LTS: estimated as the difference in temperature between 1.5 km altitude and surface) during RTE (red) and clear (blue) days, respectively. Data from all the stations across the eastern coastal box as depicted in Fig.3a is used. Although the clear days cases distribution peak maximizes around ~-7K, the RTE cases skewed towards -4K suggesting enhancement in LTS, i.e. temperature stratification, favouring the "dome effect".

[Figure]

*Figure 4. (a) The probability distribution of difference in the temperature between 1.5 km and surface during RTE (red) and clear (blue) days, obtained across all the station over the eastern coast. Probability distribution of ABL-H across (b) the north stations (Kolkata and Bhubaneshwar and Vizag, (c) Chennai and Karaikal obtained during RTE (red) and clear(blue) days.*

The enhancement in the LTS can have stronger impact on the ABL-H. Figure 4b and c shows the probability distribution of ABL-H determined from the radiosonde observations across the eastern coast. To enhance the sample size, we divided our domain box into two regions and combined the representative IMD stations for this analysis. The regions are a) north (Kolkata, Bhubaneshwar and Vizag), and b) south (Chennai and Karaikal). As expected, the mode of distribution of evening ABL-H during clear day composite varies between 1.5 – 2 km over north region and 2 -2.5 km over south region. During RTE days the ABL-H decreases significantly to 0.25-0.5 km over north region and 0.5 -1 km over south sites, suggesting the strong influence of TAL-associated LTS on suppression of ABL across the eastern coast. It is also interesting to observe a latitudinal shift in the peak occurrence of ABL-H during the RTE days. While the north regions shows a strong peak in the distribution, it broadens over the south regions, viz., Chennai and Karaikal. As discussed earlier, the pollutants to the southern regions are mostly transported

from the northern most region. These regions are strongly influenced by the frequent aerosol accumulation prevalent to the strong westerly winds over the IGP, potentially inhibiting the ABL development. The discussions are included in the revised manuscript.

**Minor Comments:**

Comment: Line 100: The statement "AOD over land at 1 km" mention including Ocean as well.

**Reply: Corrected.**

Comment: Line 146: The phrase "eastern coast within ±5° longitudes" is ambiguous. Clarify whether this refers to a specific latitude/longitude or the entire eastern coast. Define the latitude/longitude range considered in this study.

**Reply: The region of interest is portrayed in Fig.1 and incorporated in the revised manuscript.**

Comment: Line 155: The authors state "during these months is evident," but do not specify which months are being referenced.

**Reply: It is corrected as "December to March"**

Comment: Line 205: The acronym TAL should be introduced here, rather than in Line 209, for better readability and context.

**Reply: Added "Transported Aerosol Layer (TAL)" at this line.**

Comment: Lines 156–157: Specify the latitude/longitude range averaged to derive the AOD values for RTE days (0.42 ± 0.08) and clear-sky days (0.23 ± 0.06).

**Reply: It is the mean AOD observed over the eastern coastal box, portrayed in the Fig.1b.**

Comment: Line 239: Replace "during the dry season" with "during the winter season" for consistency and clarity.

**Reply: Corrected as "during the winter season"**

**References**

**Feng, Y., V. R. Kotamarthi, R. Coulter, C. Zhao, and M. Cadeddu (2016), Radiative and thermodynamic responses to aerosol extinction profiles during the pre-monsoon month over South Asia, *Atmos. Chem. Phys.*, 16(1), 247–264, doi:10.5194/acp-16-247-2016.**

**Thomas, A., Sarangi, C., & Kanawade, V. P. (2019). Recent Increase in Winter Hazy Days over Central India and the Arabian Sea. *Scientific Reports*, 9(1), 1-10. https://doi.org/10.1038/s41598-019-53630-3**

**Thomas, A., Kanawade, V. P., Sarangi, C., & Srivastava, A. K. (2021). Effect of COVID-19 shutdown on aerosol direct radiative forcing over the Indo-Gangetic Plain outflow region of the**

Bay of Bengal. *Science of The Total Environment*, *782*, 146918. https://doi.org/10.1016/j.scitotenv.2021.146918

Yu, H., Remer, L. A., Chin, M., Bian, H., Tan, Q., Yuan, T., & Zhang, Y. (2012). Aerosols from Overseas Rival Domestic Emissions over North America. *Science*. https://doi.org/1217576

Stohl, A. (2006). Characteristics of atmospheric transport into the Arctic troposphere. *Journal of Geophysical Research: Atmospheres*, *111*(D11). https://doi.org/10.1029/2005JD006888

---

## Author Response (AR2)

**Response to Handling Editor:**

Dear Authors:

Thank you for revising the MS. The revision is adequate, but requires some technical corrections. Some corrections are shown below, but not complete. Please go through the MS and correct them.

Many thanks and best,

J Kuttippurath

**Reply: Thank you very much for your positive evaluation of our revised manuscript and for guiding it toward acceptance. We appreciate your feedback and have carefully addressed the technical corrections you highlighted.**

**In addition, we have thoroughly reviewed the entire manuscript to ensure consistency and have corrected other technical issues to the best of our ability.**

L9-10: it sounds that the air mass advection occurs only from the "polluted" region

**Reply: Omitted the phrase "heavily pollution" from the sentence.**

L13: enable

**Reply: Corrected**

L14: delete "layer", use "aerosols"

**Reply: Corrected**

L63-64: Sorry to refer our work here, but this study ( DOI: 10.1016/j.heliyon.2023.e17940 ) is very relevant and clearly demonstrates your point. Transport of aerosols from the IGP and northern India to the BoB region, through trajectory and potential source contribution function (PSCF) analysis.

**Reply: Thank you for highlighting this relevant study. We appreciate the suggestion and agree that the work by Kuttippurath, J. et al., 2023, provides important supporting evidence for the aerosol transport mechanisms discussed in our manuscript. We have now cited this study appropriately in the revised version.**

L113, 129-130: space between latitude and longitude indicators (e.g. 22. 65 N). Degree symbol is also missing. Check other places too.

**Reply: Checked and corrected.**

L132: "6.9 m" and in other places

**Reply: Checked and corrected.**

L164, 219: "50 m" and in other places

**Reply: Checked and corrected.**

L174: Sometimes you use 70th, but in some other place 70th (e.g. L266). Be consistent in using them.

**Reply: Corrected everywhere.**

L205: Sometimes you use Indo-Gangetic PLAIN, and sometimes "PLAINS" with "S". Be consistent in using this.

**Reply: Corrected as Indo-Gangetic Plain (IGP) everywhere.**

L231: 2 days, … up to 5 days

**Reply: Corrected**

L257, 259 and other places: Supplementary Fig. S1. Delete the "figure"

**Reply: Corrected everywhere.**

L336: ". Barbaro et al. (2014) suggest that .."

**Reply: Corrected.**

L337: "Similarly, Wang et al. (2015) suggest .. "

**Reply: Corrected.**

L341: the same correction as for L337

**Reply: Corrected.**

L401: "2 km"

**Reply: Corrected.**